# CD47-SIRPα Signaling Induces Epithelial-Mesenchymal Transition and Cancer Stemness and Links to a Poor Prognosis in Patients with Oral Squamous Cell Carcinoma

**DOI:** 10.3390/cells8121658

**Published:** 2019-12-17

**Authors:** Shin Pai, Oluwaseun Adebayo Bamodu, Yen-Kuang Lin, Chun-Shu Lin, Pei-Yi Chu, Ming-Hsien Chien, Liang-Shun Wang, Michael Hsiao, Chi-Tai Yeh, Jo-Ting Tsai

**Affiliations:** 1Graduate Institute of Clinical Medicine, College of Medicine, Taipei Medical University, Taipei City 110, Taiwan; einfachweiss@gmail.com (S.P.); 12061@s.tmu.edu.tw (C.-S.L.); mhchien1976@gmail.com (M.-H.C.); wangls72269@yahoo.com.tw (L.-S.W.); 2Department of Oral & Maxillofacial Surgery, Saint Martin de Porres Hospital, Chaiyi City 600, Taiwan; 3Department of Medical Research and Education, Taipei Medical University—Shuang Ho Hospital, New Taipei City 235, Taiwan; 16625@s.tmu.edu.tw; 4Department of Hematology and Oncology, Cancer Center, Taipei Medical University—Shuang Ho Hospital, New Taipei City 235, Taiwan; 5Biostatistics Center, Taipei Medical University, Taipei City 110, Taiwan; robbinlin@tmu.edu.tw; 6Department of Radiation Oncology, Tri-Service General Hospital, National Defense Medical Center, Taipei City 114, Taiwan; 7Department of Pathology, Faculty of Medicine, Fu Jen Catholic University, New Taipei City 242, Taiwan; stardrboypo5210@gmail.com; 8Department of Thoracic Surgery, Taipei Medical University—Shuang Ho Hospital, New Taipei City 235, Taiwan; 9Genomics Research Center, Academia Sinica, Taipei City 115, Taiwan; mhsiao@gate.sinica.edu.tw; 10Department of Medical Laboratory Science and Biotechnology, Yuanpei University of Medical Technology, Hsinchu City 300, Taiwan; 11Department of Radiology, School of Medicine, College of Medicine, Taipei Medical University, Taipei City 110, Taiwan; 12Department of Radiology, Taipei Medical University—Shuang Ho Hospital, New Taipei City 235, Taiwan

**Keywords:** chemoradiation, oral cancer stem cells, CD47, radiotherapy techniques, radioresistance

## Abstract

Background: Oral squamous cell carcinoma (OSCC), with high mortality rates, is one of the most diagnosed head and neck cancers. Epithelial-to-mesenchymal transition (EMT) and the generation of cancer stem cells (CSCs) are two keys for therapy-resistance, relapse, and distant metastasis. Accumulating evidence indicates that aberrantly expressed cluster of differentiation (CD)47 is associated with cell-death evasion and metastasis; however, the role of CD47 in the generation of CSCs in OSCC is not clear. Methods: We investigated the functional roles of CD47 in OSCC cell lines SAS, TW2.6, HSC-3, and FaDu using the bioinformatics approach, immunoblotting, immunofluorescence staining, and assays for cellular migration, invasion, colony, and orosphere formation, as well as radiosensitivity. Results: We demonstrated increased expression of CD47 in OSCC patients was associated with an estimated poorly survival disadvantage (*p* = 0.0391) and positively correlated with the expression of pluripotency factors. Silencing CD47 significantly suppressed cell viability and orosphere formation, accompanied by a downregulated expression of CD133, SRY-Box transcription factor 2 (SOX2), octamer-binding transcription factor 4 (OCT4), and c-Myc. In addition, CD47-silenced OSCC cells showed reduced EMT, migration, and clonogenicity reflected by increased E-cadherin and decreased vimentin, Slug, Snail, and N-cadherin expression. Conclusion: Of therapeutic relevance, CD47 knockdown enhanced the anti-OSCC effect of radiotherapy. Collectively, we showed an increased CD47 expression promoted the generation of CSCs and malignant OSCC phenotypes. Silencing CD47, in combination with radiation, could provide an alternative and improved therapeutic efficacy for OSCC patients.

## 1. Introduction

Oral cancer is one of the most diagnosed malignancies, with a global annual incidence of 300,373 cases [1]. Regardless of diagnostic and therapeutic progress, from an estimated 145,353 deaths for both sexes in 2012, the oral cancer-related mortality rate continues to rise, with a predicted global mortality rate of 67.1% (n = 242,886) by 2035 because of demographic changes, thus making it one of the main causes of cancer-related deaths globally [1,2]. Constituting over 90% of all oral cancer histological subtypes, the oral squamous cell carcinoma (OSCC) is notably highly aggressive, often non-responsive to common anticancer therapy, and associated with early relapse and poor prognosis [1,2].

Currently, the treatment of choice remains surgery, radiation therapy, and/or chemotherapy, which consists of cisplatin (CDDP), docetaxel (DTX), and 5-fluorouracil (5-FU), especially for advanced-stage (III and IV) malignancies; however, this is often associated with increased risk of severe drug-induced adverse effects and toxicities, high rates of treatment failure and disease recurrence, as well as low median survival rates for OSCC patients [3], with an estimated 5-year survival rate that is consistently below 50% over the last 30 years [4]. In fact, despite the documented therapeutic relevance of radiotherapy for the reduction of OSCC tumor size and oral function preservation, it is common for irradiated OSCC patients to develop early loco-regional disease relapse, thus, further contributing to the dismal prognosis [5]. Putting together, these inform the need for the discovery of novel actionable molecular target or development of novel therapeutic agents with better efficacy.

The last two decades have been characterized by accumulating evidence, implicating cancer stem cells (CSCs) in the initiation and sustenance of tumor, resistance to chemo- and/or radiation therapy, disease recurrence, and metastases [6,7].

Previously, we demonstrated that the direct targeting of CSCs or so-called tumor-initiating cells (TICs) in OSCC by gene loss of function [8] or therapeutic targeting [9] inhibited the malignant and metastatic traits of OSCC cells, eliminated their CSCs-like properties, including enhanced clonogenicity, orosphere formation, and self-renewal, as well as improved their sensitivity to chemo- and/or radiation therapy, thus, highlighting the therapeutic significance of preferentially targeting the CSCs pool as an efficacious anti-OSCC strategy for mitigating the menace of radioresistance, cancer recurrence, and metastasis, while improving survival rates in patients with OSCC. Corollary to the above implication of CSCs in cancer metastases and resistance to anticancer therapy, as well as accruing association of the aberrant expression of epithelial-to-mesenchymal transition (EMT)-inducing transcription factors with cancer stemness, the last decade has been characterized by increased documentation of existent reciprocal link between CSCs and EMT, with CSCs exhibiting EMT phenotypes, and EMT being relevant to the acquisition and maintenance of stem cell-like traits and sufficient to confer same traits on differentiated non-cancerous and cancerous cells [10,11,12]. This CSCs-EMT reciprocity, mirroring a pathological positive feedback loop, plays a critical role in enhanced chemoresistance, radiotherapy resistance, disease progression, recurrence, and poor prognosis [10,11,12,13].

The ubiquitously-expressed transmembrane glycoprotein CD47, also known as integrin-associated signal transducer or protein (IAST or IAP), belongs to the immunoglobulin (Ig) superfamily and is an antigenic surface determinant protein that provides a “self” or “do not eat” signal by interacting with the N-terminus of its ligand, signal regulatory protein alpha (SIRPα) on immune cells to suppress or inhibit macrophage phagocytosis [14]. In fact, by interacting with integrins from the β1, β2, and β3 families, CD47 induces heterotrimeric G-protein signaling, thus modulating leukocyte adhesion, cell motility, and phagocytosis. It has been recently demonstrated that CD47 is upregulated in leukemia cells and circulating hematopoietic stem cells to avoid pre-mobilization and intra-mobilization phagocytosis by macrophages, thus, lending credence to the known role of CD47 as a facilitator of immune-surveillance evasion by cancer cells [15]. However, while the nefarious role of CD47 in tumorigenesis, therapy evasion, and poor prognosis is increasingly being demystified, the therapeutic promise of CD47-mediated CSCs-targeting remains poorly understood and largely unexplored in OSCC; thus, in the present study, we explored the probable role of loss of CD47 function in the attenuation of oral CSCs, EMT, and radiosensitivity.

In this study, we investigated and provided evidence for the feasibility of a CD47-mediated attenuation of OSCC cell viability, suppression of OSCC-SCs-like attributes and associated pluripotency, deregulation of the constitutive CSCs-EMT loop in OSCC, and enhancement of OSCC cell sensitivity to radiation therapy.

## 2. Materials and Methods

### 2.1. Patient Samples

We evaluated the prognostic relevance of CD47 expression in a cohort of OSCC patients (n = 71) aged between 29 and 72, with a median age of 48 years, who had undergone definitive treatment with curative intent in National Defense Medical Center from October 2000 to March 2013. The present study was reviewed and approved by the institutional review board (TSGHIRB 2–102-05–125), and all participants provided written informed consent. Patient distribution based on American Joint Committee on Cancer (AJCC) staging, 7th edition, was as follows - 9 stage I disease (13%), 22 stage II (31%), 13 stage III (18%), and 27 stage IV (38%), as well as normal non-tumor (2), mild dysplasia (1), moderate dysplasia (1), and severe dysplasia (3) from peritumoral tissue. Location-wise, buccal mucosa (n = 39, 51.3%) was the most common affected site, followed by tongue (n = 25, 32.9%), gingival (n = 7, 9.2%), and others (n = 5, 6.6%) [palate (n= 2), maxilla (n = 1), lip (n = 1), and tonsil (n = 1)]. None of the specimens had received radiation or a chemotherapeutic regimen. The clinicopathological characteristic of our cohort is shown in Table 1.

### 2.2. Immunohistochemical (IHC) Staining

For the CD47 IHC staining analysis of OSCC (n = 71) and non-tumor oral tissues (n = 7), 1% bovine serum albumin (BSA) was used for blocking the formalin-fixed paraffin-embedded (FFPE) tissue sections before they were incubated with CD47 antibody (Santa Cruz Biotechnology Inc, Santa Cruz, CA, USA) at 4 °C overnight. The sections were then incubated with goat anti-mouse IgG (Cell Signaling Technology Inc., Danvers, MA, USA) for 1 h. Tissue staining was scored by two independent pathologists. The staining index (SI) was calculated based on the formula.
***SI = [percentage of positively stained OSCC cells] × [staining intensity]***
where percentage of positively stained tumor cells was graded as: 0 (0%), 1 (<10%), 2 (10–35%), 3 (35–70%), and 4 (>70%); and staining intensity graded as: 0 (no staining), 1 (weak), 2 (moderate), and 3 (strong). Positive results were defined as buffy granules in the cell membrane and cytoplasm. Scores of intensities were given 1, 2, and 3 depending on the color intensity as weak, moderate, strong staining. The quick score (Q score) is defined as the percentage of staining cells (%) multiplied by a score of intensity. The membrane and cytoplasm scores were calculated in each. The total Q score was given as membrane score plus cytoplasm score. 

### 2.3. Reagents

Gibco® RPMI 1640, trypsin/EDTA, dimethyl sulfoxide (DMSO), phosphate-buffered saline (PBS), sulforhodamine B (SRB) medium, acetic acid, and TRIS base were also purchased from Sigma Aldrich Co. (St. Louis, MO, USA).

### 2.4. Cell Lines and Culture

The human OSCC cell lines HSC-3 and FaDu were obtained from the American Type Culture Collection (ATCC. Manassas, VA., USA), while the SAS and TW2.6 cells were kindly provided by Prof. Hsiao Michael (Genomic Research Center, Academia Sinica). Both cell lines were cultured in Dulbecco’s modified Eagle’s medium (DMEM, Invitrogen Life Technologies, Carlsbad, CA, USA), supplemented with 10% fetal bovine serum (FBS) and 1% penicillin/streptomycin (Invitrogen, Life Technologies, Carlsbad, CA, USA) and incubated at 37 °C in 5% humidified CO2 incubator. The OSCC cells were passaged at 98% confluence, and the medium was changed every 72 h before exposure to 5 Gy–15 Gy of radiotherapy. 

### 2.5. shRNA Transfection of OSCC Cell Lines

The SAS, TW2.6, HSC-3, or FaDu cells were then transfected with shRNA specifically targeting CD47 or control/scramble shRNA purchased from Shanghai GenePharma Co., Ltd (Shanghai, China). The shRNA sequences were as follows: shCD47-1 - 5’-CGTCACAGGCAGGACCCACTGCCCA-3’; shCD47-2 - 5’-CCACAGATG TACAAGGGATGACCACAGTGTCATT-3’; and scramble shCD47 - 5’-CGTGACAGCCACGACCGACTGCGCA-3’. The OSCC cells were transfected with packed and harvested shRNA-coding lentiviruses following the manufacturer’s instruction. Briefly, 5 × 10^4^ SAS or TW2.6 cells plated in 24-well plates were primed with 8 μg/mL hexadimethrine bromide, and viral particles were added to the culture medium at a multiplicity of infection (MOI) of 6. After transfection for 12 h, the virus-containing medium was replaced with fresh culture medium. The selection of viable OSCC cells stably transfected with shCD47 was performed with 2 μg/mL puromycin the next day, and the surviving colonies were expanded for further experiments.

### 2.6. Orosphere Formation and Self-Renewal Assay

For generation of orospheres, the OSCC cells were seeded at 5 × 10^4^ cells/well in 6-well non-adherent plates (Corning Inc., Corning, NY, USA) in serum-free DMEM/F12 medium (11330057; Thermo Fisher Scientific Inc, Bartlesville, OK, USA.) supplemented with basic fibroblast growth factor (bFGF) (20 ng/mL, Invitrogen, Carlsbad, CA, USA), B27 supplement (Invitrogen, Carlsbad, CA, USA), 5 μg/mL insulin (91077C; Sigma Aldrich Co. (St. Louis, MO, USA)), and epidermal growth factor (EGF) (20 ng/mL, Millipore, Bedford, MA, USA). The cells were then incubated at 37 °C in a 5% humidified CO_2_ incubator for 12–15 days, and the formed orospheres were observed and counted using an inverted phase-contrast microscope. After 12 days of culture, primary orospheres consisting of ≥50 μm were counted, and images were taken under the microscope. Secondary orospheres were generated by dissociating primary orospheres using the trypsinization method, and the dissociated primary orospheres were then pipetted through a 22G needle to obtain a single-cell suspension (Thermo Fisher Scientific Inc, Bartlesville, OK, USA). After dissociation of the primary orospheres, cell seeding was done as for primary tumorspheres. After 12–15 days of culture, secondary orospheres consisting of ≥50 µm were counted, and the images were taken under the microscope.

### 2.7. Colony Formation Assay

WT or shCD47 SAS, TW2.6, HSC-3, or FaDu cells were plated in triplicate at 2 × 10^4^ cells per well in 6-well plates and cultured for 12 days. After the colonies formed attained the cluster size of ≥50 cells, they were washed with PBS two times, fixed with 95% methanol for 15 min, and then stained with crystal violet at room temperature for 15 min. This was followed by colony visualization and counting under the microscope. The size and number of colonies formed were estimated with the ChemiDoc-XRS imager from the QuantityOne software package (Bio-Rad, Hercules, CA, USA).

### 2.8. Radiation and Cell Viability Assay

3 × 10^4^ WT or shCD47 SAS, TW2.6, HSC-3, or FaDu cells were seeded and cultivated per plate in triplicate in 96-well plates. After 24 h, both adherent and floating cells were harvested, baseline cell-count was carried out, then triplicate wells were irradiated with 5 Gy–15 Gy or left without irradiation for another 24 h. The irradiated or unexposed OSCC cells were then harvested, fixed with 10% trichloroacetic acid (TCA), washed with ddH_2_O, and stained with 0.4% SRB (w/v) in 1% acetic acid. The unbound SRB dye was carefully washed off with 1% acetic acid several times, and then plated air-dried. The bound SRB dye was solubilized in 10 mM Trizma base, and absorbance was read using a microplate reader at 570 nm wavelength. Variation in cell numbers was relative to the baseline number of viable cells before irradiation. 

### 2.9. Western Blot Analysis

For western blotting, cells were lysed using RIPA buffer (Cell Signaling Technology Inc., Danvers, MA, USA) with a cocktail of protease inhibitors (Sigma, St. Louis, MO, USA). Ten micrograms of protein samples were separated using 10% SDS-PAGE electrophoresis and transferred to polyvinylidene fluoride (PVDF) membranes using the Bio-Rad Mini-Protein electro-transfer system (Bio-Rad Laboratories Inc, CA, USA). Then, non-specific binding was blocked by incubating the membranes in 5% skimmed milk in Tris-buffered saline with 0.1% Tween 20 (TBST) for 1 h at room temperature, and then the membranes were incubated overnight at 4 °C with the antibodies against CD47 (1:1000, B6H12, sc-12730, Santa Cruz Biotechnology Inc, Santa Cruz, CA, USA), OCT4 (1:1000, C-10, sc-5279, Santa Cruz), SOX2 (1:1000, A-5, sc-365964, Santa Cruz), CD133 (1:1000, MAB4399-1, EMD Millipore, Temecula, CA, USA), vimentin (1:1000, D21H3, #5741, Cell Signaling Technology Inc., Danvers, MA, USA), Slug (1:1000, C19G7, #9585, Cell Signaling Technology Inc., Danvers, MA, USA), Snail (1:1000, C15D3, #3879, Cell Signaling Technology Inc., Danvers, MA, USA), N-cadherin (1:1000, D4R1H, #13116, Cell Signaling Technology Inc., Danvers, MA, USA), E-cadherin (1:1000, 24E10, #3195, Cell Signaling Technology Inc., Danvers, MA, USA), and GAPDH (1:500, G-9, sc-365062, Santa Cruz), listed in Appendix A. The membranes were later incubated with the corresponding horseradish peroxidase (HRP)-conjugated anti-rabbit IgG or anti-mouse IgG secondary antibodies (Cell Signaling Technology Inc., Danvers, MA, USA) for 1 h at room temperature and washed with PBS four times. Protein band detection was done by the enhanced chemiluminescence (ECL) detection system (Thermo Fisher Scientific Inc., Waltham, MA, USA), and protein band quantification was done using ImageJ v. 1.46 (https://imagej.nih.gov/ij/). Target protein expression was normalized to that of GAPDH, and assays were repeated four times in triplicates.

### 2.10. Wound Healing Migration Assay

For the wound-healing migration assay, OSCC cells were seeded in 6-well plates and cultured to 100% confluence. At full confluence, a 1 mm wound was made along the median axis of the well using a yellow pipette tip. Cell migration into the wound area was then observed at 0, 3, 6, and 12 h time-points under the microscope. The assay was performed three times in triplicates.

### 2.11. Matrigel Invasion and Migration Assay

Cell invasion assays were performed in Boyden chambers (pore size = 8 μm) with the upper side of the filter covered with 0.2% Matrigel diluted in DMEM. The 1 × 10^4^ irradiated or un-irradiated WT, shCD47-1, or shCD47-2 OSCC cells in serum-free culture medium were plated in the upper chambers, while the lower chambers contained complete culture medium with 10% FBS as a chemoattractant. After overnight incubation, the un-invaded OSCC cells on the upper side of the filter were carefully removed with a cotton bud, while the invaded cells on the lower side of the membrane were washed, fixed in 95% ethanol and then stained with 10% Giemsa dye. The number of invaded cells was counted in 4 randomly selected fields of each membrane and averaged to obtain a representative number of the invaded cells. For the Boyden chamber migration assay, the porous membranes without the Matrigel were used.

### 2.12. Immunofluorescence Staining

For the immunofluorescence analysis, the OSCC cells were plated in 6-well chamber slides (Nunc™, Thermo Fisher Scientific Inc., Waltham, MA, USA) for 24 h, fixed in 2% paraformaldehyde at room temperature for 10 min, permeabilized with 0.1% Triton X-100 in 0.01 M PBS (pH 7.4) containing 0.2% BSA, air-dried, and rehydrated in PBS. The cells were then incubated with antibody against CD47, Oct4, Sox2, c-Myc, vimentin, or E-cadherin diluted 1:200 in PBS containing 3% normal goat serum at room temperature for 2 h, followed by incubation with anti-rabbit IgG fluorescein isothiocyanate (FITC)-conjugated secondary antibody (Jackson ImmunoResearch Lab. Inc., West Grove, PA, USA). The cells were allowed to rest at room temperature for 1 h, washed in PBS, and mounted using Vectashield mounting medium while counterstaining with 4′,6-diamidino-2-phenylindole (DAPI, D3571, Molecular Probes, Life Technologies Co., Carlsbad, CA, USA). Images were captured using a Zeiss Axiophot (Carl Zeiss Co. Ltd., Hsinchu City, Taiwan) fluorescence microscope, and the microphotographs were analyzed using ImageJ software v. 1.46 (https://imagej.nih.gov/ij/).

### 2.13. Statistical Analysis

Each experiment was performed at least 3 times in triplicates. All statistical analyses were carried out using IBM SPSS Statistics for Windows, Version 25.0 (Released 2017; Armonk, NY: IBM Corp. USA). All data represent means ± standard deviation (SD). Comparison between two groups was estimated using the 2-sided Student’s *t*-test, while the one-way analysis of variance (ANOVA) was used for comparison between 3 or more groups. The association between the differential expression of CD47 and overall survival (OS) in patients with OSCC was determined using univariate Cox proportional regression of covariates, including the age, gender, AJCC stage, pathological grade, local recurrence, and lymph node involvement. Variables for which *p* < 0.05 were identified as significantly associated with prognosis, and Cox multivariate analysis was subsequently performed for these variables. Hazard ratios (HRs) and 95% confidence intervals (CIs) for multivariate analyses were computed using the Cox proportional hazards regression. *p*-value <0.05 was considered statistically significant.

## 3. Results

### 3.1. CD47 Is Aberrantly Expressed in Human Oral Squamous Cell Carcinoma and Influence Survival Rate

To understand the role of CD47 in OSCC, we performed computational analyses of CD47 RNAseq expression profile in 33 different cancer types (n = 9736 tumors), including head and neck squamous cell carcinoma (HNSCC) matched with corresponding normal samples from the cancer genome atlas (TCGA) and genotype-tissue expression (GTEx) (n = 8587 non-tumors) datasets, using the analysis of variance for the evaluation of differential expression. Results of these analyses indicated that CD47 was overexpressed in tumors compared to adjacent non-tumor oral tissues and was mostly expressed in HNSCC tissues with ~ 125 transcripts per million (TPM) after the ovarian (OV, ~198 TPM) and lung adenocarcinoma (LUAD, ~195 TPM) (Figure 1A and Appendix A). Furthermore, statistical analyses of OSCC cohort data (consisting of the oral tongue, buccal mucosa, alveolar, oropharynx, tonsils, floor of the mouth, and base of the tongue) downloaded from the UCSC Xena functional genomics browser (https://xena.ucsc.edu/) was used to further characterize CD47 expression in patients with OSCC; our results indicated that CD47 was overexpressed in the keratinizing, non-keratinizing, and ‘not otherwise specified’ (NOS) squamous cell carcinoma of the oral cavity, which constituted over 75% of HNSCC (Figure 1B). Further computational analyses of OSCC in TCGA cohort (n = 412; *p* = 0.0009) showed that compared to expression in the ‘normal’ oral epithelium (n = 32), CD47 was significantly more expressed in the OSCC tissue samples (n = 380) (Figure 1C).

We also demonstrated using downloaded and reanalyzed malignant OSCC data from the TCGA HNSCC cohort that high CD47 expression conferred a significant survival disadvantage in OSCC patients with high CD47 expression, compared to those with low CD47 expression (*p* = 0.0391; Figure 1D).

### 3.2. The Aberrant Expression of CD47 in Human Oral Squamous Cell Carcinoma Tissue Positively Correlates with Disease Progression

Furthermore, consistent with earlier data, compared to the normal or dysplastic tissues, results of our immunohistochemical staining showed varying degrees of positive CD47 staining in all 71 OSCC cases; of which, 87.5% were membranous, 10.9% cytoplasmic, and 1.6% perinuclear staining. A strong positive correlation between enhanced CD47 protein expression and disease progression or tumor stage was established (Figure 2A). Interestingly, while we observed no apparent CD47 expression in ‘normal’ non-dysplastic tissues, we observed a graduated mild positive CD47 expression in the ‘non-tumor’ mild to severely dysplastic tissues, moderate expression of CD47 in the early stage (I, II) carcinoma (*p* < 0.05 vs. normal or mild dysplasia), and strong CD47 staining in the late stage (III and IV) group (*p* < 0.001 vs. normal or mild dysplasia), especially in the cytomembranous region (Figure 2A–C). These findings were corroborated by the univariate proportional hazard analyses of our clinicopathological variables (Table 2), which demonstrated that similar to disease progression parameters, such as lymph node (LN) involvement (pN) (Fisher’s exact test, *p* = 0.001), presence of local recurrence (Fisher’s exact test, *p* = 0.003), and late American Joint Committee on Cancer (AJCC) stage (Fisher’s exact test, *p* = 0.002), high CD47 expression was strongly associated with worse survival ((HR (95%CI) = 6.83 (1.72 – 18.09), *p* = 0.01)) and multivariate analyses (Table 2), indicating that enhanced CD47 expression was also an independent prognosticator of poor clinical outcome *cum* higher risk of disease-specific death ((multivariate: HR(95%CI) = 5.18 (0.73 – 12.64), *p* = 0.019)), akin to local recurrence (Fisher’s exact test, *p =* 0.031) and AJCC stage (Fisher’s exact test, *p =* 0.048). Together these data did indicate the active role of CD47 in OSCC carcinogenesis and poor prognosis.

### 3.3. CD47 Modulates the Cancer Stem Cell-Like Phenotype and Self-Renewal in Oral Squamous Cell Carcinoma Cells

Having established a correlation between OSCC carcinogenesis, poor prognosis, and CD47 expression, we further sought to unravel the underlying mechanism. Against the background that c-Myc is a master regulator of cancer transcriptome, epigenome, and immune privilege [16,17], and that OSCC SCs are characterized by ectopic expression of CD133 and pluripotency master regulators, including SOX2, OCT4, and NANOG [18,19], we probed for probable correlation and/or functional association between CD47, OCT4, SOX2, CD133, and c-MYC. Preliminary analyses of the human OSCC genome U133A array of the Toruner HNSCC cohort (n = 20) showed concomitant upregulation of CD47 (1.55-fold, *p* = 0.00003), SOX2 (1.40-fold, *p* = 0.01), and CD133 (1.13-fold, *p* = 0.06) in the OSCC tissues, compared to normal tissue group (Figure 3A). Further, shRNA-mediated silencing of CD47 expression generated shCD47-1 and shCD47-2 in SAS (knockdown efficiency: 92% and 90%, respectively) and TW2.6 (knockdown efficiency: 93% and 80%, respectively) cell lines (Figure 3B). In similar experiments, our western blot analyses revealed that shCD47 elicited significant co-suppression of CD47, SOX2, OCT4, and CD133 in orosphere-derived SAS, HSC-3, and FaDu cells transfected with shCD47 (Figure 3C and Appendix A). These Western blot results were corroborated by immunofluorescence (IFC) staining, showing a concurrent remarkable reduction in orosphere size and expression levels of CD47, OCT4, c-MYC, and SOX2 protein in the shCD47-1 and shCD47-2 TW2.6 cells, compared with their WT counterpart (Figure 3D). These results were suggestive of a functional association between CD47 and CSCs regulators in OSCC cells, as well as highlighted the probable modulatory role of CD47 on the CSCs-like phenotype of OSCC cells. However, understanding that at the core of OSCC chemoresistance, metastasis, and recurrence, lies the capability of CSCs to regenerate all components of the primary or original tumor and drive cancer aggression [6,7,10,11,12,13], we examined if and how shCD47 affects the orosphere-forming capability of single-cell solutions derived from dissociated primary orospheres and cultured in stem cell media (recapitulating OSCC-SC self-renewal). We demonstrated that shCD47-1 and shCD47-2 markedly inhibited the self-renewal potential of the SAS (shCD47-1: 69.1% inhibition, *p* < 0.001; shCD47-2: 88.3% inhibition, *p* < 0.001) or TW2.6 (shCD47-1: 71% inhibition, *p* < 0.05; shCD47-2: 73.5% inhibition, *p* < 0.01) primary orospheres (Figure 3E,F). In addition, similar to the primary orospheres, we showed that the shCD47-induced reduction in the number and sizes of SAS or TW2.6 orospheres positively correlated with significant inhibition of the nuclear expression of pluripotency transcription factors, SOX2 and OCT4 (Figure 3G). These data indicated that shCD47 efficaciously suppressed the CSCs-like phenotype, including the expansion and self-renewal of the OSCC-SC population in vitro.

### 3.4. Downregulation of CD47 Attenuate the EMT and Migration Capacity of OSCC Cells

Understanding that apart from their involvement in cell growth, evasion of cell death, therapy resistance, and poor prognosis, CSCs are also implicated in the EMT, migration/invasion, and EMT-induced metastasis [20], we next investigated if and how alteration in CD47 expression affect the motility of OSCC cells. Using wound-healing migration assays, we demonstrated that the migratory ability of the shCD47-1 and shCD47-2 SAS cells were significantly lesser than that of the WT control SAS cells at 3 h, 6 h, and 9 h (Figure 4A). Similarly, clonogenicity assay showed that compared to the SAS WT cells, the ability to form colonies was significantly reduced in the shCD47-1 (51%, *p* < 0.05) and shCD47-2 (63%, *p* < 0.01) SAS cells (Figure 4B). Furthermore, we examined for likely correlation between altered CD47 expression and the expression of EMT markers, such as vimentin, Slug, Snail, N-cadherin, and E-cadherin, in human OSCC, using western blot. Transfecting the SAS, HSC-3, or FaDu cells with shCD47 upregulated the expression of E-cadherin and downregulated the expression of mesenchymal markers N-cadherin, vimentin, Slug, and Snail (Figure 4C and Appendix A). In corroboratory assays, we also observed that the loss of CD47 in SAS (upper panel) and TW2.6 (lower panel) cells led to the loss of the characteristic OSCC spindle-shaped/fibroblastoid mesenchymal morphology, acquisition of round/oval epithelial and loose cell-to-cell contact (Figure 4D). These results indicated that the downregulation of CD47 in OSCC cells attenuated their EMT and migration potential.

### 3.5. CD47 Modulates the Expression and Subcellular Localization of Mesenchymal and Epithelial Factors in OSCC

In parallel assays, immunofluorescence (IFC) staining showed that compared to expression in the negative control shCD47 scramble cells, the SAS orospheres (SAS Sp) cells harbored enhanced expression of vimentin (Figure 5A) but decreased expression of E-cadherin (Figure 5B). We also demonstrated that in the shCD47 cells, the observed reduced expression of CD47 positively correlated with marked orosphere disintegration, significant reduction in orosphere size, loss of cytoplasmic and nuclear co-localization of CD47 with vimentin, and enhanced cytoplasmic/cytomembranous co-localization of CD47 with E-cadherin (Figure 5A,B). Using the R2: genomics analysis and visualization platform (https://hgserver1.amc.nl/), we carried out a bivariate analysis of the transcript expression profiles of CD47, vimentin (VIM), or E-cadherin (CDH1) in the Roepman OSCC cohort (n = 220). Results of our analyses further validated earlier demonstrated direct and inverse correlation of CD47 with VIM and CDH1, respectively, with the significance of correlation r-value = 0.081 *p*-value = 0.23 T-value = 1.198 for CD47 versus VIM, and r-value = −0.061 *p*-value = 0.37 T-value = −0.898 for CD47 versus CDH1, with both sharing same degrees of freedom = 218 (Figure 5C,D). In addition, for better appreciation and visualization of the place of CD47 in the interplay and functional interaction between CSCs and EMT, we generated an association network of the interaction between CD47 and molecular moieties involved in stem cell development, stem cell differentiation, stem cell maintenance, positive regulation of cell migration, and positive regulation of cell motility, based on physical interactions (38.77%), genetic interactions (0.95%), co-expression (17.82%), shared protein domains (3.89%), pathway (1.67%), and molecular prediction (36.91%) (Figure 5E). These data indicated that CD47 not only interacted with but also modulated the expression and subcellular localization of mesenchymal and epithelial factors in OSCC.

### 3.6. Suppression of CD47 Expression Enhances the Sensitivity of OSCC-SCs to Radiation Therapy 

Understanding the critical role of radiotherapy as a vital treatment modality in OSCC that facilitates oral tumor size reduction and oral function preservation, we next examined the effect of altered CD47 expression on the viability and/or proliferation of the OSCC cell lines, SAS and TW2.6, using the radiotherapy-based cell viability assay. Twenty-four hours post-transfection with shCD47-1, shCD47-2, or negative control scramble shCD47, SAS cells were exposed to 5 Gy–15 Gy radiation. The combination of *shCD47* and irradiation induced significantly greater cell-death in comparison to the radiation only group, as evidenced by ~52% vs. 46%, 24% vs. 19%, and 28% vs. 20% reduction in the viability of shCD47-1 vs. shCD47-2 transfected SAS cells treated with 5 Gy, 10 Gy, and 15 Gy, respectively (Figure 6A,B, also see Appendix A). The inhibitory concentration at which irradiation reduced the cell viability by 50% at 8.59 Gy, 1.06 Gy, and 1.71 Gy for WT, shCD47-1, and shCD47-2 SAS cells, respectively, is shown in Figure 6B. Along the same line, we evaluated the probable enhanced effect of combining molecular attenuation of CD47 with low-dose radiotherapy. Our results showed that compared to migration of the un-irradiated SAS WT, significantly lesser migratory potential was exhibited by the 5 Gy irradiated shCD47-1 and shCD47-2-transfected cells (~5.2-fold, *p* < 0.001), un-irradiated shCD47 alone (~2.7-fold, *p* < 0.001), and 5 Gy radiation alone SAS WT (1.8-fold, *p* < 0.01) (Figure 6C). Similarly, a 3.5-fold (*p* < 0.01), 3-fold (*p* < 0.01), or 12-fold (*p* < 0.001) reduction in the number of invaded un-irradiated shCD47-transfected, 5 Gy radiation alone, or 5 Gy irradiated shCD47-transfected cells, respectively, was observed (Figure 6D and Appendix A). Consistent with the CD47-CSCs-EMT positive feedback loop already alluded to above, we also demonstrated that compared to the orospheres generated from SAS WT shCD47 cells, fewer and smaller orospheres were formed by SAS or HSC-3 WT cells irradiated with 5 Gy alone, cells transfected with shCD47-1 alone, and 5 Gy-irradiated cells harboring shCD47-1, in decreasing order of magnitude (Figure 6E and Appendix A). Furthermore, in combination with shCD47, exposure to 5 Gy radiation reduced the number of colonies formed by ~8.69–11.1-fold (*p* < 0.001) in comparison to un-irradiated SAS WT (Figure 6F, also see Appendix A). Our data did indicate that the observed resultant enhanced the anticancer effect of radiation therapy combined with shCD47 in OSCC-SCs was primarily synergistic and highly effective. These results demonstrated that molecular attenuation of shCD47 as a therapeutic strategy abrogated the CSCs-related radio-resistance of OSCC cells.

## 4. Discussion

Despite advances made in the diagnostic approach and therapeutic strategies, OSCC remains a clinical challenge with great financial and emotional implications for patients and caregivers alike. The failure of conventional anticancer therapy in OSCC is increasingly being associated with the presence and activities of CSCs, where these OSCC-SCs are constitutively resistant to chemotherapeutic agents and radiation therapy. The last two decades have been characterized by heightened interest in and increased exploration of the role of CSCs in the resistance or reduced sensitivity of OSCC cells to standard chemotherapeutics or radiation therapy, with most research focused on the discovery of new therapeutic targets and/or development of novel anticancer drugs that preferentially target CSCs-enrichment in oral cancer cells [5,6,7,8,9,21]. 

In this present study, we hypothesized and provided evidence that the ”don’t eat me” signal—CD47—modulated radiosensitivity by CSCs regulation and EMT deactivation in OSCC cells. We demonstrated that CD47 was aberrantly expressed in human OSCC and influenced the survival rate of patients with OSCC (Figure 1 and Figure 2; Table 2). This finding corroborated the recently published work of Ye X., et al., who showed the overexpression of CD47 in Tca8113, Cal-27, and SCC-9 OSCC cell lines and week expression of same in normal oral keratinocytes. Consistent with our data showing that high CD47 expression conferred ~20% survival disadvantage and positively correlated with disease stage progression, they also suggested that CD47 might serve as a reliable predictive biomarker for oral pre-cancer and cancer progression, thus hinting on its probable role as an important molecular target for designing novel anticancer therapeutics for OSCC patients [22]. Also demonstrated to be ectopically expressed in several other cancer types, strong evidence abounds implicating CD47 in the pathogenesis and progression of malignancies by its inherent ability to inhibit the phagocytosis of transformed or malignant cells [23,24,25]. This anti-phagocytosis potential aligning with the cancer cells’ escape from immune surveillance, evasion of cell death, and sustenance of chronic proliferation sequel to enhanced mitogenic and pluripotency signals is evocative of the Hanahan and Weinberg’s hallmarks of cancer [26] and reminiscent of our current understanding of CSCs biology and activities [27]. Our understanding is that cancer cells are neither homogeneous nor solitary bio-entities; they interact with their microenvironment with the emission of inhibitory and activating cues, such as the mitogenic and angiogenic signals, respectively, to facilitate tumor progress [26,27]. The heterogeneous cell phenotype, tumor-initiating potential, perpetuity of OSCC cells, and the poorly chartered territory of OSCC-SCs eradication is now at the forefront of oral cancer research. The current paradigm is that a small subset of cancer cells, which in the context of this present work is termed OSCC-SCs, confers the unique self-renewal and cancer regeneration abilities on OSCC cells [27]. 

In search of OSCC-relevant actionable molecular targets, our work for the first time, to the best of our knowledge, demonstrated that CD47 modulated the CSCs-like phenotype in OSCC, as evidenced by concomitant upregulation of known pluripotency transcription factors and stemness markers with CD47 expression, and co-suppression of SOX2, OCT4, c-MYC, and CD133 protein expression following shRNA-mediated downregulation of CD47 expression in wild-type adherent and anchorage-independent orospheres (Figure 3). As already alluded above, CSCs within the OSCC tumor nests co-express OCT4, SOX2, NANOG, phosphorylated STAT3 (pSTAT3), CD133, CD44, and c-MYC [18]. Our result is of clinical relevance since, in the CSCs model, the net loss of cancer cells due to programmed or spontaneous cell death elicits a short-term reduction in population size, but over the long-term, counter-intuitively promotes tumor growth, as previously quiescent OSCC-SCs are reactivated, re-enter the cell cycle, proliferate, and repopulate the tumor bulk. Thus, the significance of efficient therapeutic targeting of OSCC-SCs is by the genetic ablation of CD47 expression and/or activity, as demonstrated in our work. Our findings are corroborated by recently demonstrated induction of macrophage phagocytosis of lung cancer cells and lung CSCs, as well as the inhibition of lung CSCs-induced tumor growth in immune-deficient mice xenograft models by blocking CD47 function with anti-CD47 antibodies [28]. Similarly, enhanced phagocytosis of CD47-rich CD133+ ovarian TICs, which are relatively resistant to current anticancer treatment, is triggered by treatment with anti-CD47 mAb or CD47 knockdown [29]. We also demonstrated that the downregulation of CD47 induced the loss of the mesenchymal (fibroblast-like) phenotype and acquisition of an epithelial (ovoid-shaped) phenotype, attenuating the EMT and migration capacity of OSCC cells (Figure 4). We posit that by facilitating cytoskeleton remodeling, cell elongation, and loss of cell-cell/cell-basal membrane adhesion, CD47 induces enhanced migration and invasion, resistance to chemoradiotherapy, and evasion of cell death, which are characteristics of EMT in OSCC; upon completion of the transition, the cells degrade the underlying extracellular matrix (ECM) and commence dispersion to secondary anatomic sites. The disruption of this CD47-induced phenotypic transition would be consistent with our demonstrated shCD47-induced reversal of cadherin switching with resultant E-cadherin/N-cadherin ratio greater than 1, suppression of E-cadherin master regulators--Snail and Slug [30], as well as the intermediate filament, vimentin, which is associated with enhanced invasiveness, higher prevalence of lymph node involvement, disease recurrence, and poor prognosis [30,31]. Aside from demonstrating that CD47 modulates the expression and subcellular localization of mesenchymal and epithelial factors in OSCC, we also provided evidence, at least in part, that CD47 was a master regulator of stem cell development, differentiation, and maintenance, as well as positive regulator of cell migration and cell motility (Figure 5). These findings are particularly insightful and therapeutically-relevant, especially as the “migrating, self-renewing and symmetrically-dividing CSCs shape the primary tumor, and are also exclusively capable of distant seeding, whereas the majority of non-stem cancer cells (that can be frequently detected as circulating tumor cells) are intrinsically only able to form dormant micrometastases” [27]. To the best of our understanding of the OSCC SC-EMT, while it is difficult to delineate clearly whether it is the undifferentiated or differentiated OSCC cells responsible for observed EMT phenotype in our study, we would allude to the recent comprehensive overview of related theme, in which Kim et al. suggested that “early, undifferentiated cells with mesenchymal phenotype are characterized by a shift from E-cadherin expression to N-cadherin expression along with the expression of Snails, vimentin and metalloproteases”, and that these “early undifferentiated cells with a mesenchymal phenotype retain the expression of several totipotent transcription factors (e.g., Oct4 and Nanog), which indicates that these cells can adopt a mesenchymal phenotype without losing their pluripotency” [32]. In light of this, we posit a regulatory role for CD47 signaling at the interphase between the CSCs and EMT phenotype of OSCC cells.

By inference, extrinsic perturbations, such as CD47 blockage/knockdown and cytotoxic treatments, including radiation therapy, may mold oral tumor by selective targeting of the aggressive OSCC cells, including OSCC-SCs, which subsequently facilitate malignant growth; thus, any efficacious therapy must eradicate OSCC-SCs; however, there is mounting evidence, showing these cells are intrinsically less or in-sensitive to current OSCC anticancer therapy. Thus, interestingly, having shown the existence of a positive correlation of CD47 expression with CSC, EMT, and metastatic phenotypes, as well as an inverse correlation with OSCC patients’ overall survival, cell death; we finally demonstrated that the suppression of CD47 expression enhanced the sensitivity of OSCC-SCs to radiation therapy, as evidenced by the remarkable synergistic effect of concurrent CD47 knockdown and radiotherapy on cell viability, migration, invasiveness, clonogenic, and orospheric survival (Figure 6). This has clinical significance since radiotherapy is a common and very vital component of the multidisciplinary treatment for patients with OSCC, especially those with unresectable oral cavity tumors, cases where surgery is technically improbable for the early-stage disease with high risk of cosmetic or functional defect, high operative risk secondary to co-morbidity or subpar performance status [33,34].

Human papillomavirus (HPV) infection has been implicated in over 25% of HNSCCs. In fact, many of the HPV positive HNSCCs are oropharynx cancer, including tongue base and tonsil [35,36]. Clinically, the HPV status of patients is being touted as a predictor of treatment response and survival rate [3,37,38]. It has been suggested that the better prognosis of HPV(+) HNSCCs involves the host immune system, especially as immune cells, which were originally suppressed in the tumor microenvironment, have been shown to be re-activated through the crosstalk of immunogenic signals between tumorous cells and immune cells by exosomes [38]. Tumors release exosomes containing a lot of antigens. In contrast to HPV(−) tumor, exosomes released by HPV (+) tumors do exhibit prolong immunogenic interaction with immune cells due to the regulation of the expression and/or activity of CD47 on the membrane of cancerous cells. The membrane protein CD47 functions as a “don’t eat me” signal and thus limits the clearance of cancerous cells by circulating monocytes [38,39]. Radiation has been shown to induce anti-tumor immune response [40] and associated with a dose-dependent decrease in the surface expression of CD47 on HPV(+) HNSCC cells, resulting in improved clearance of tumorous cell, in vitro and in vivo [39]. 

Our initial analysis of the TCGA-HNSCC cohort (n = 604) gene expression profile dataset showed it contained limited number of the HPV positive HNSCC as sorted by FISH or p16 testing; most of the tumor sites were located at the tongue base and oropharynx, which were excluded in our original TCGA analysis since our study focus was oral cancer instead. In addition, the expression level of CD47 in these samples was low or null. These findings suggested that the therapeutic potential of targeting CD47 was likely independent of the HPV status of the patients with OSCC. The mechanism underlying the response of CD47 to radiation in OSCC remained relatively underexplored, thus, necessitating a further investigation of the use of CD47 as a potential therapeutic target in anti-OSCC treatment.

As with most studies of this nature, this present study had several limitations within which the findings presented herein should be interpreted carefully. This study was a retrospective, single-center investigation of the selected biomarker in a relatively small cohort of Taiwanese OSCC patients (n = 71), limiting the ‘generalizability’ of the findings reported, especially to other ethno-geographics and raises the question of probable predisposition to selection biases.

In conclusion, as depicted in Figure 7, we demonstrated that CD47 knockdown alone or combined with radiation therapy significantly inhibited the survival and/or proliferation of OSCC-SCs. The CD47 knockdown of shCD47 also suppressed orosphere formation, reduced colony formation, and enhanced radiosensitivity in OSCC by dysregulation of our proposed CD47-CSCs-EMT signaling loop. The findings of our present study lent credence to the role of CD47 as a putative actionable therapeutic target with high anticancer efficacy and laid another brick on the foundation for further exploration of the clinical feasibility and therapeutic application of ‘CD47 ablation’ as a novel strategy for improving therapeutic outcome in oral cancer clinics.

## Figures and Tables

**Figure 1 cells-08-01658-f001:**
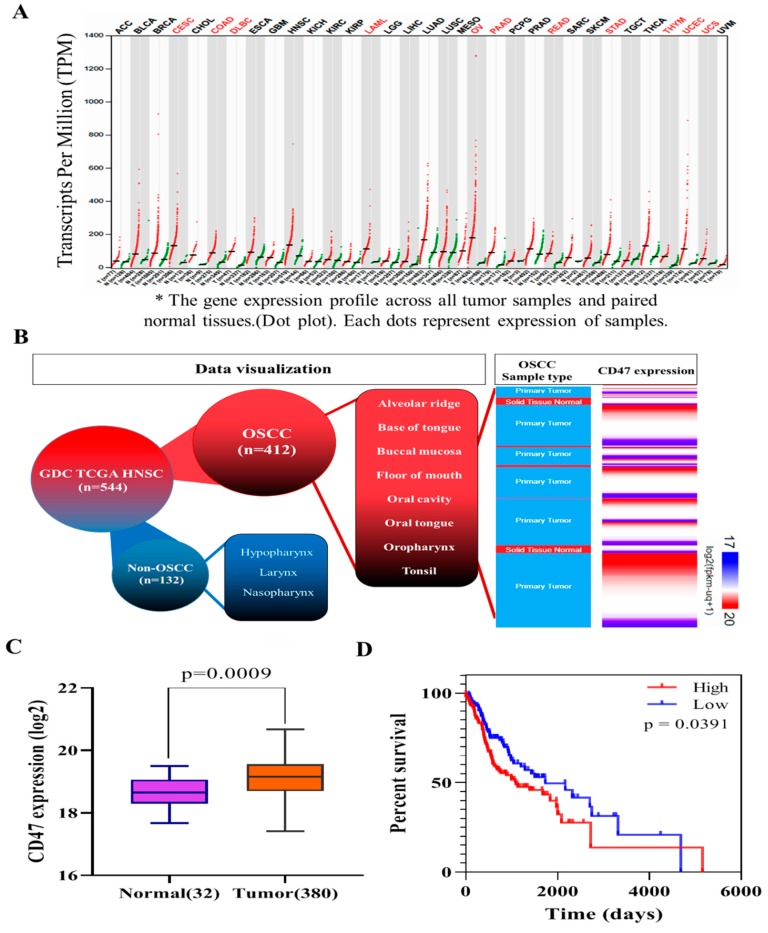
CD47 is aberrantly expressed in human oral squamous cell carcinoma and influence survival rate. (**A**) CD47 transcript expression profile across TCGA and GTEx paired normal-tumor tissue cohort. (**B**) The expression of CD47 in downloaded data for OSCC based on morphology, anatomic site, and sample type from the Genomic Data Commons-The Cancer Genome Atlas (GDC TGCA) HNSCC dataset. (**C**) Differential expression of CD47 in normal oral and cancer tissues in TCGA OSCC cohort (n = 412; *p* = 0.0009). (**D**) Kaplan–Meier curves showing the effect of low and high CD47 expression on the overall survival of the TGCA malignant OSCC cohort. OSCC: oral squamous cell carcinoma; GTEx: genotype-tissue expression; HNSCC: head and neck squamous cell carcinoma; GDC: genome data commons; TCGA: the cancer genome atlas.

**Figure 2 cells-08-01658-f002:**
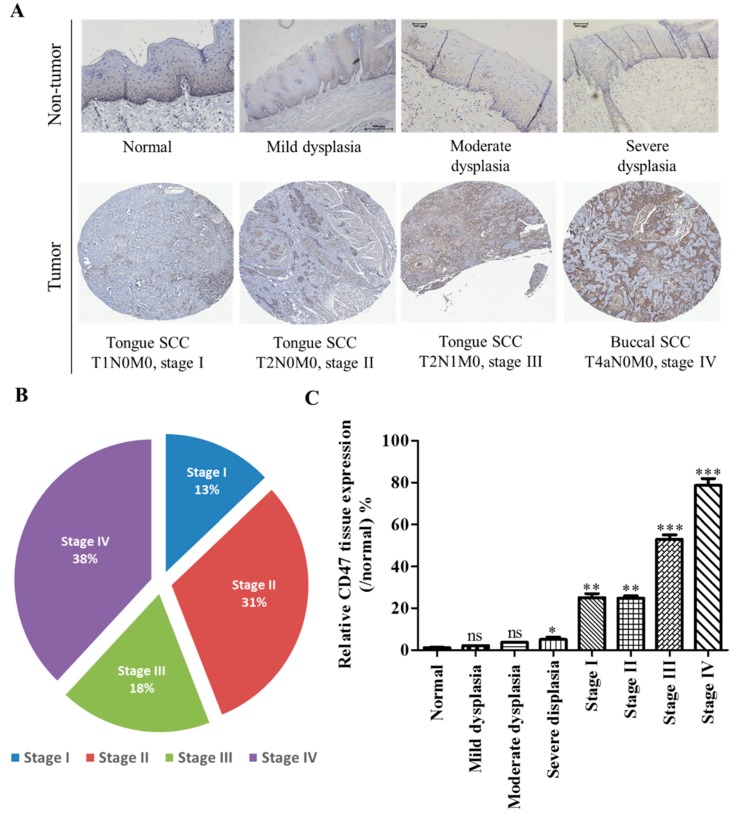
The aberrant expression of CD47 in oral squamous cell carcinoma positively correlates with disease progression. (**A**) Representative immunohistochemistry staining of CD47 in human normal oral and OSCC tissues. (**B**) Pie chart showing the distribution of patients in our OSCC cohort (n = 71) based on histological types. (**C**) Graphical representation of the histology-specific relative expression of CD47 in tissue samples from our OSCC cohort. CD47 tissue expression is relative to that in the normal group. ns, not significant; * *p* < 0.05, ** *p* < 0.01, *** *p* < 0.001.

**Figure 3 cells-08-01658-f003:**
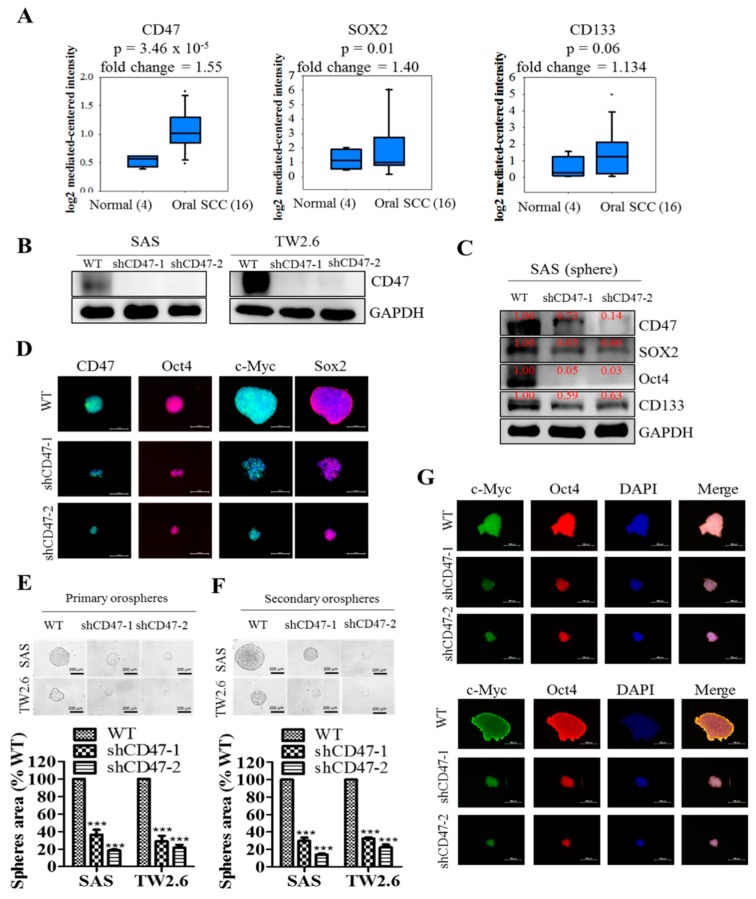
CD47 modulates the cancer stem cell-like phenotype in oral squamous cell carcinoma cells. (**A**) Box and whiskers chart showing the correlative differential expression of CD47 (upper), SOX2 (middle), and CD133 (lower) mRNA from analyses of the human OSCC genome U133A array from the Toruner head-neck cohort, n = 20. (**B**) Knockdown efficiency of shCD47-1 and shCD47-2 on the protein expression of CD47 in SAS and TW2.6 cell lines shown by western blot analysis. (**C**) Effect of CD47 knockdown on the expression level of CD47, SOX2, OCT4, and CD133 proteins in SAS Sp, shCD47-1, or shCD47-2 cells shown by western blot analysis. GAPDH served as a loading control. (**D**) Immunofluorescent staining showing the effect of shCD47 on the expression of CD47, OCT4, c-Myc, and SOX2 proteins in spheres formed by TW2.6 cells. TW2.6 and SAS cells transfected with shCD47-1 or shCD47-2 exhibited decreased orosphere size (left) and number (right) in both (**E**) primary and (**F**) secondary generation orospheres. (**G**) shCD47 attenuated OCT4 and SOX2 expression and inhibited their nuclear co-localization in TW2.6- or SAS-derived orospheres, as shown by immunofluorescent (IFC) staining. All assays are representative of experiments performed four times in triplicates. WT, wild type; Sp, orosphere; blue stain = DAPI, nuclear staining. All data are representative of experiment carried out four times in triplicate and are expressed as mean ± S.D. * *p* < 0.05, ** *p* < 0.01, *** *p* < 0.001.

**Figure 4 cells-08-01658-f004:**
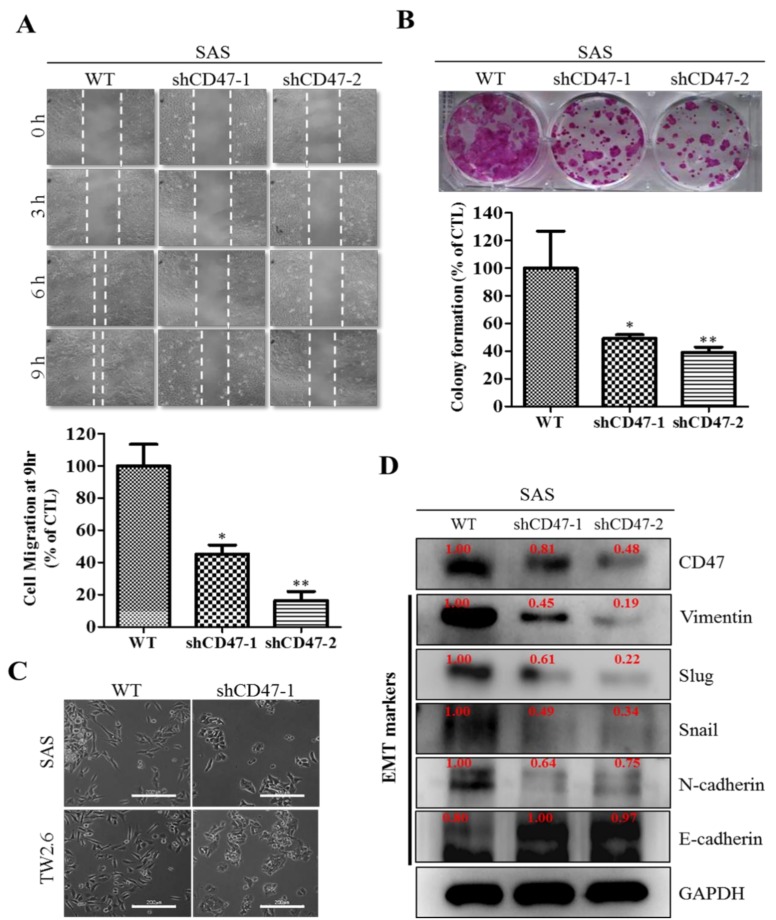
Downregulation of CD47 attenuates the EMT (epithelial-to-mesenchymal transition) and migration capacity of OSCC cells. (**A**) Representative image of scratch-wound migration assay shows the effect of shCD47 on the motility of SAS cells at 0, 3, 6, and 9-h time points (upper), and quantitative bar chart of the migrating cell fronts at indicated time points (lower). (**B**) Representative images of colony formed by WT, shCD47-1, or shCD47-2 transfected SAS cells in the culture plate using crystal violet solution and quantification of visible cells. (**C**) The inhibitory effect of shCD47 on the expression of CD47, vimentin, Slug, Snail, N-cadherin, and E-cadherin in SAS cells, as demonstrated by western blot analyses. (**D**) Photo-image, showing the fibroid/spindle shape of CD47-expressing WT cells, while shCD47 led to the loss of mesenchymal phenotype in SAS (upper) and TW2.6 (lower) cells. WT, wild type; GAPDH served as a loading control. All data are representative of experiment carried out four times in triplicate and are expressed as mean ± S.D. * *p* < 0.05, ** *p* < 0.01, *** *p* < 0.001.

**Figure 5 cells-08-01658-f005:**
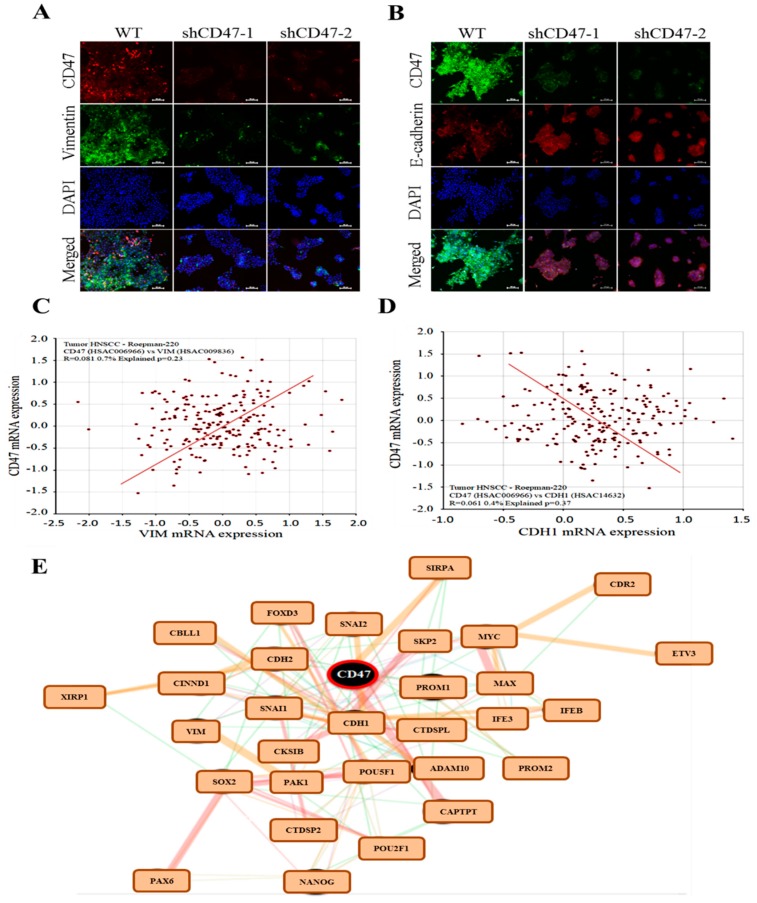
CD47 modulates the expression and subcellular localization of mesenchymal and epithelial factors in OSCC. (**A**,**B**) Immunofluorescent staining images, showing the expression of CD47, vimentin, and E-cadherin in SAS WT, and shCD47-1 or shCD47-2 cells. Representative dot plot of the correlative expression of (**C**) CD47 mRNA versus vimentin mRNA or (**D**) CD47 mRNA versus CDH1 mRNA in OSCC patients from the Roepman cohort, n = 220 using the R2 online genomic analysis and visualization software. (**E**) The associative network of the interaction between CD47 and molecular moieties involved in stem cell development, stem cell differentiation, stem cell maintenance, positive regulation of cell migration, and positive regulation of cell motility. Sp, orosphere; DAPI, nuclear staining.

**Figure 6 cells-08-01658-f006:**
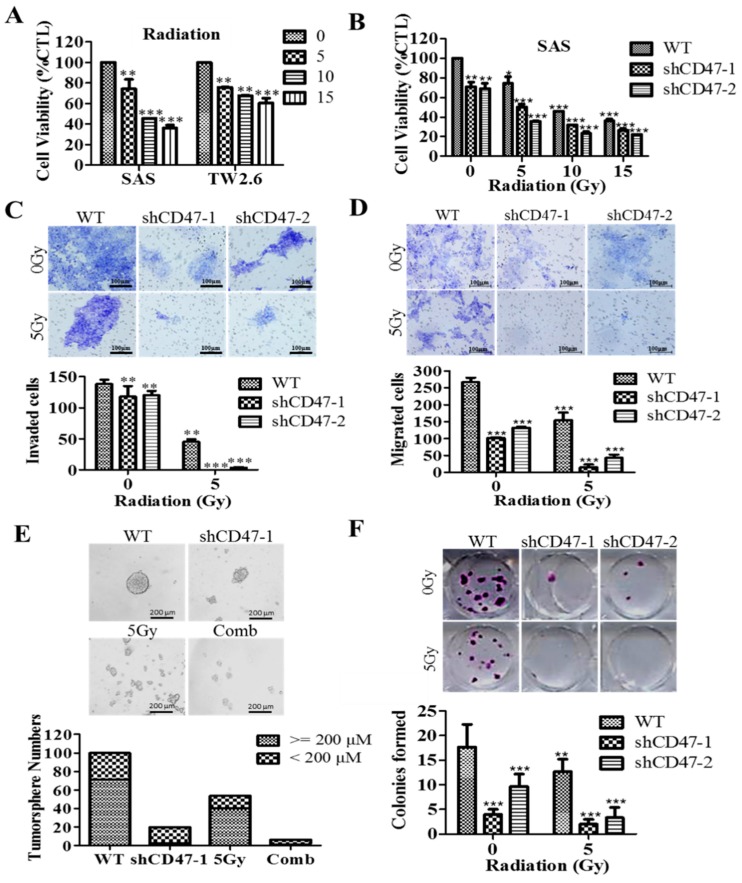
Suppression of CD47 expression enhances the sensitivity of OSCC-SCs to radiation therapy. (**A**) Bar chart of the inhibitory effect of exposure to 0 Gy–15 Gy radiation on the viability of SAS or TW2.6 cells. (**B**) shCD47 with or without 5 Gy–15 Gy radiation decreased the viability of SAS cells dose-dependently. (**C**) Transwell migration assay images show reduced migration in 5 Gy-exposed shCD47 SAS cells, compared to their SAS WT counterparts. (**D**) Transwell invasion assay images show reduced invasion in 5 Gy-exposed shCD47 SAS cells, compared to their SAS WT counterparts. (**E**) shCD47-transfected cells yielded smaller and fewer tumorspheres compared to their WT or shCD47 scramble counterparts. (**F**) shCD47-1 or shCD47-2 SAS cells formed fewer colonies when exposed to 5 Gy, compared to the SAS WT cells. Data represent mean ± SD from three independent experiments performed in triplicates. * *p* < 0.05, ** *p* < 0.01, and *** *p* < 0.001.

**Figure 7 cells-08-01658-f007:**
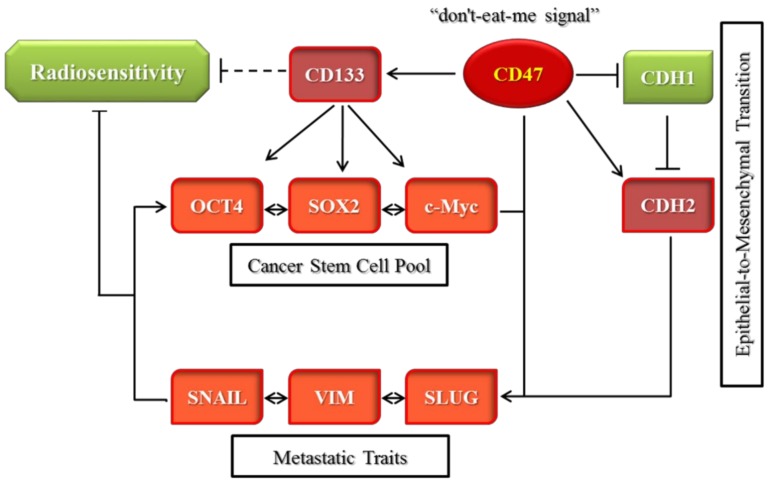
Schema showing CD47-related molecular network and how the suppression of the ”don’t-eat-me” signal CD47 enhances radiosensitivity by downregulating cancer stem cells-associated pluripotency factors and deactivating epithelial-to-mesenchymal transition in oral squamous cell carcinoma cells.

**Table 1 cells-08-01658-t001:** Clinicopathological characteristics of 71 patients with OSCC (oral squamous cell carcinoma).

Variables	*N*
Age [median (range)]	48 (29–72)
Sex (M/F)	62/9
Total	71
**Follow-up**	
Range (days)	161–3645
**AJCC* staging**	
Early (I+II)	31
Late (III+IV)	40
**Tumor**	
T1	13
T2	18
T3	21
T4	19
**Nodes**	
N (−)	34
N (+)	37

*AJCC, American Joint Committee on Cancer 7th ed.

**Table 2 cells-08-01658-t002:** Univariate and multivariate Cox regression analysis of clinicopathological parameters and CD47 expression for overall survival in patients with OSCC.

Clinicopathological Variables	Univariate			Multivariate#		
	HR	95%CI	*p*-Value	HR	95%CI	*p*-Value
**Age, years** (≦mean *vs.* > mean)	0.97	0.41–2.64	0.58			
**Sex** (Male *vs.* Female)	0.55	0.03–1.78	0.42			
**Tumor** (T1 + T2 *vs.* T3 + T4)	1.27	0.52–3.59	0.36			
**Nodes** (N1 + N2 + N3 *vs.* N0)	3.25	1.27–5.16	0.001	1.73	0.51–3.19	0.817
**Pathological grade** (grades 2 + 3 *vs.* grade 1)	1.35	0.67–3.02	0.28			
**Local recurrence** (Yes *vs.* No)	3.82	1.16–7.68	0.003	4.26	1.98–9.28	0.031
**CD47 expression** (high *vs.* low)	6.83	1.72–18.09	0.01	5.18	0.73–12.64	0.019
**AJCC* Staging** (III + IV *vs.* I + II)	7.67	1.90–15.37	0.002	3.42	1.09–9.85	0.048

*AJCC, American Joint Committee on Cancer; HR, hazard ratio; CI, confidence interval; Bold value represent significant *p*-value (*p* < 0.05); # Covariates with *p*-value < 0.05 in the univariate analysis were included in the multivariate model.

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
