# Peer review of "CD47-SIRPα Signaling Induces Epithelial-Mesenchymal Transition and Cancer Stemness and Links to a Poor Prognosis in Patients with Oral Squamous Cell Carcinoma"

_cells, 2019, doi:10.3390/cells8121658_

Round 1
Reviewer 1 Report
In this revised version of the manuscript the authors provide an important addition to their stem cell study in adding Figure 3E and Figure 3F. Using self-renewal essay in primary and secondary orospheres they confirmed their previous hypothesis that CD47 modulates CSC maintenance. Indeed, inhibiting CD47 expression results in a reduction of spheres size.
With these new experiments the authors significantly increased the value of the manuscript that might now be considered for publication.
Author Response
Answers to the comments:
Point-by-point responses to reviewer’s comments:
We would like to thank the reviewer for the thorough reading of our manuscript as well as their valuable comments. We have followed their comments closely and feel that they have further strengthened the manuscript. Below are our point-by-point responses.
Editor Decision
Options to upload a new version of your manuscript will be available once all the reviewer comments have been replied to.
Q1: Reviewer #1
In this revised version of the manuscript the authors provide an important addition to their stem cell study in adding Figure 3E and Figure 3F. Using self-renewal essay in primary and secondary orospheres they confirmed their previous hypothesis that CD47 modulates CSC maintenance. Indeed, inhibiting CD47 expression results in a reduction of spheres size. With these new experiments the authors significantly increased the value of the manuscript that might now be considered for publication.
A1: We thank the reviewer for taking time to review our work once again and for this encouraging comment.

Reviewer 2 Report
Dear authors,
I have read and revised with interest the manuscript:"CD47-SIRPα Signaling Induces Epithelial-Mesenchymal Transition, Cancer Stemness and Links to a Poor Prognosis in Patients with Oral Squamous Cell Carcinoma". The manuscript is of scientific interest and I congratulate for your hard work. However, in my opinion, it still needs some improvements first to be considered for publication on your prestigious journal.
Major Revisions
- On Table 1, authors state to have included 71 OSCC patients, but on Figure 2 they report than 10 of these cases were precancer disease (5 K in situ, 3 severe dysplasia...). Please clarify, this is very important for the inclusion of this study in future meta-analysis.
- It is unclear which AJCC staging edition was applied to the patients included in the analysis (7th or 8th edition?)
- No info about survival analysis is reported on "Statistical Analysis", the section should be improved. I suggest authors to follow the REMARK guidelines.
- Since the number of patients included is low, authors performed bioinformatic analysis. This is a good option to increase the power of findings. However, the bioinformatic analysis has been performed on the whole cohort of head and neck squamous cell carcinoma. Authors should focus only on OSCC, hence I suggest to download both clinical and CD47 gene expression data from Xena or cBIOportal and perform all the statistical analysis according to the REMARK guidelines.
Minor Revisions
- On page 4 lines 5-6 and 8, write in the correct manner the incidence of oral cancer (300, 373; 145, 353 and 242, 886 cases), remove the comma and replace it with a dot.
- Figure 2B appear much more as a Pie Chart than a Venn diagram
(Reporting Recommendations for Tumor Marker Prognostic Studies (REMARK): explanation and elaboration. Altman DG, McShane LM, Sauerbrei W, Taube SE. PLoS Med. 2012;9(5):e1001216. doi: 10.1371/journal.pmed.1001216.)
Author Response

(The authors gave the same response as above.)

Editor Decision
Options to upload a new version of your manuscript will be available once all the reviewer comments have been replied to.
Q1: Reviewer #2: I have read and revised with interest the manuscript:"CD47-SIRPα Signaling Induces Epithelial-Mesenchymal Transition, Cancer Stemness and Links to a Poor Prognosis in Patients with Oral Squamous Cell Carcinoma". The manuscript is of scientific interest and I congratulate for your hard work. However, in my opinion, it still needs some improvements first to be considered for publication on your prestigious journal.
A1: We thank the reviewer for taking time to review our work once again, for the encouraging comments, and for the suggestion given once again to help improve our work.
Q2: Reviewer #2: On Table 1, authors state to have included 71 OSCC patients, but on Figure 2 they report than 10 of these cases were precancer disease (5 K in situ, 3 severe dysplasia...). Please clarify, this is very important for the inclusion of this study in future meta-analysis.
A1: We sincerely thank the reviewer for pointing out this computation error. We now carefully reassessed and corrected likely discrepancies in patient samples number. Please see our updated Table 1 in page 25 and Figure 2 in page 28.
Table 1. Clinicopathological characteristics of 76 patients with OSCC.
Figure 2. The aberrant expression of CD47 in oral squamous cell carcinoma positively correlates with disease progression. (A) Representative immunohistochemistry staining of CD47 in human OSCC tissues. (B) Pie chart showing the distribution of patients in our cohort (n=76) based on histological types. (C) Graphical representation of the histology-specific relative expression of CD47 in tissue samples from our cohort. CD47 tissue expression is relative to that in the normal group. ns, not significant; *p<0.05, **p<0.01, ***p<0.001.
Please see our revised Materials and Methods section, page 6, lines 2-19.
Patient Samples
We evaluated the prognostic relevance of CD47 expression in a cohort of OSCC patients (n=76) aged between 29 and 72, with median age of 48 years, who had undergone definitive treatment with curative intent in National Defense Medical Center from October 2000 to March 2013. The present study was reviewed and approved by the institutional review board (TSGHIRB 2–102-05–125 and TMU-JIRB Form056/20190306), and all participants provided written informed consents. Patient distribution based on American Joint Committee on Cancer (AJCC) staging, 7th edition, was as follows, 5 patients with carcinoma in situ (6.6%), 9 had stage I disease (11.8%), 22 with stage II (28.9%), 13 with stage III (17.1%), and 27with stage IV(35.5%), as well as normal non-tumor (2), mild dysplasia (1), moderate dysplasia (1) and severe dysplasia (3) from peritumoral tissue. Location-wise, buccal mucosa (n= 39, 51.3%) was the most common affected site, followed by tongue (n= 25, 32.9%), gingival (n= 7, 9.2%), and others (n=5, 6.6%) [palate (n= 2), maxilla (n= 1), lip (n= 1) and tonsil (n= 1)]. None of the specimens had received radiation or chemotherapeutic regimen. The clinicopathological characteristic of our cohort is shown in Table 1.
Also kindly see our revised Results section, page 11, Lines 31-43 to page 12, line 1-14.
The aberrant expression of CD47 in human oral squamous cell carcinoma tissue positively correlates with disease progression
Furthermore, consistent with earlier data, results of our immunohistochemical staining showed positive CD47 staining in 64 of the 76 (84.2%) cases; of these 87.5% of these were membranous, 10.9% cytoplasmic, and 1.6% perinuclear staining. A strong positive correlation between enhanced CD47 protein expression and disease progression or tumor stage was established (Figure 2A). Interestingly, while we observed no apparent CD47 expression in ‘normal’ non-dysplastic tissues, we observed a graduated mild positive CD47 expression in the ‘non tumor’ mild to severely dysplastic tissues, moderate expression of CD47 in the oral carcinoma in situ (stage 0) and early stage (I, II) carcinoma (p<0.05 vs. normal or mild dysplasia), and strong CD47 staining in the late stage (III) group (p<0.001 vs. normal or mild dysplasia), especially in the cytomembranous region (Figure 2A-C). These findings were corroborated by the univariate proportional hazard analyses of our clinicopathological variables (Table 2) which demonstrated that similar to disease progression parameters such as lymph node (LN) involvement (pN) (Fisher’s exact test, p=0.001), presence of local recurrence (Fisher’s exact test, p=0.003), and late American Joint Committee on Cancer (AJCC) stage (Fisher’s exact test, p=0.002), high CD47 expression is strongly associated to worse survival ((HR (95%CI) = 6.83 (1.72 – 18.09), p=0.01)), and multivariate analyses (Table 2) indicating that enhanced CD47 expression is also an independent prognosticator of poor clinical outcome cum higher risk of disease-specific death ((multivariate: HR(95%CI) = 5.18 (0.73 – 12.64), p=0.019)), akin to local recurrence (Fisher’s exact test, p=0.031) and AJCC stage (Fisher’s exact test, p=0.048). Together these data do indicate the active role of CD47 in OSCC carcinogenesis and poor prognosis.
Q3: Reviewer #2: It is unclear which AJCC staging edition was applied to the patients included in the analysis (7th or 8th edition?)
A1: We thank the reviewer for this important comment. We have now indicated the AJCC staging 7th edition in the revised manuscript. Please kindly see Materials and Methods section, Page 6, Lines 2-19.
Patient Samples
We evaluated the prognostic relevance of CD47 expression in a cohort of OSCC patients (n=76) aged between 29 and 72, with median age of 48 years, who had undergone definitive treatment with curative intent in National Defense Medical Center from October 2000 to March 2013. The present study was reviewed and approved by the institutional review board (TSGHIRB 2–102-05–125 and TMU-JIRB Form056/20190306), and all participants provided written informed consents. Patient distribution based on American Joint Committee on Cancer (AJCC) staging, 7th edition, was as follows, 5 patients with carcinoma in situ (6.6%), 9 had stage I disease (11.8%), 22 with stage II (28.9%), 13 with stage III (17.1%), and 27with stage IV(35.5%), as well as normal non-tumor (2), mild dysplasia (1), moderate dysplasia (1) and severe dysplasia (3) from peritumoral tissue. Location-wise, buccal mucosa (n= 39, 51.3%) was the most common affected site, followed by tongue (n= 25, 32.9%), gingival (n= 7, 9.2%), and others (n=5, 6.6%) [palate (n= 2), maxilla (n= 1), lip (n= 1) and tonsil (n= 1)]. None of the specimens had received radiation or chemotherapeutic regimen. The clinicopathological characteristic of our cohort is shown in Table 1.
Q4: Reviewer #2: No info about survival analysis is reported on "Statistical Analysis", the section should be improved. I suggest authors to follow the REMARK guidelines.
A1: We are very grateful to the reviewer for this suggestion. We have now address the statistical issue raised by the reviewer by adding some information about our survival analysis in statistical analysis. Please kindly see our revised Materials and Methods section, page 10, lines 5-20.
Statistical Analysis
Each experiment was performed at least 3 times in triplicates. All statistical analyses were carried out using IBM SPSS Statistics for Windows, Version 25.0 (Released 2017; Armonk, NY: IBM Corp. USA). All data represent means ± standard deviation (SD). Comparison between two groups was estimated using the 2-sided Student's t-test, while the one-way analysis of variance (ANOVA) was used for comparison between 3 or more groups. The association between the differential expression of CD47 and overall survival (OS) in patients with OSCC was determined using univariate Cox proportional regression of covariates including the age, gender, AJCC stage, pathological grade, local recurrence, and lymph node involvement. Variables for which p<0.05 were identified as significantly associated with prognosis, and Cox multivariate analysis was subsequently performed for these variables. Hazard ratios (HRs) and 95% confidence intervals (CIs) for multivariate analyses were computed using the Cox proportional hazards regression. P-value <0.05 was considered statistically significant.
Q5: Reviewer #2: Since the number of patients included is low, authors performed bioinformatic analysis. This is a good option to increase the power of findings. However, the bioinformatic analysis has been performed on the whole cohort of head and neck squamous cell carcinoma. Authors should focus only on OSCC, hence I suggest download both clinical and CD47 gene expression data from Xena or cBIOportal and perform all the statistical analysis according to the REMARK guidelines.
A1: We sincerely appreciate the reviewer’s comment. However, all data used and presented in the current study are indeed OSCC data from the HNSCC of TCGA or OSCC cohorts from other portals. Please kindly see our Results section, pages 11, line 2-29.
CD47 is aberrantly expressed in human oral squamous cell carcinoma and influence survival rate
To understand the role of CD47 in OSCC, we performed computational analyses of CD47 RNAseq expression profile in 33 different cancer types (n = 9 736 tumors) including head and neck squamous cell carcinoma (HNSC) matched with corresponding normal samples from the cancer genome atlas (TCGA) and genotype-tissue expression (GTEx) (n = 8, 587 non-tumors) datasets, using the analysis of variance for evaluation of differential expression. Results of these analyses indicate that CD47 was overexpressed in tumors compared to adjacent non-tumor oral tissues and was mostly expressed in HNSC tissues with ~ 125 transcripts per million (TPM) after the ovarian (OV, ~198 TPM) and lung adenocarcinoma (LUAD, ~195 TPM) (Figure 1A and Supplementary Figure S1). Using the UCSC Xena functional genomics browser (https://xena.ucsc.edu/) to further characterize CD47 expression in the genome data commons (GDC) TCGA HNSC, our histomorphological stratification analyses revealed that CD47 is overexpressed in the keratinizing, non-keratinizing or ‘not otherwise specified’ (NOS) squamous cell carcinoma of the oral cavity, constituting over 80% of the HNSC and consisting of the oral tongue, buccal mucosa, alveolar, oropharynx, tonsils, floor of the mouth, and base of the tongue, compared to other histomophological types and anatomic sites (Figure 1B). Consistent with results from our computational analyses, immunohistochemical staining of TMU-SHH OSCC cohort (n = 76) showed that compared to expression in the ‘normal’ oral epithelium (n = 2), CD47 was significantly more expressed in the OSCC tissue samples (p = 0.0017; Figure 1C). We also demonstrated using our OSCC cohort that high CD47 expression confers a significant survival disadvantage in OSCC patients, as evidenced by a 2-year 20% reduced survivability in patients with high CD47 expression, compared to those with low CD47 expression (Figure 1D).
Also kindly see our revised Results section, page 14, line 1-30.
CD47 modulates the expression and subcellular localization of mesenchymal and epithelial factors in OSCC
In parallel assays, immunofluorescence (IFC) staining showed that compared to expression in the negative control shCD47 scramble cells, the SAS orospheres (SAS Sp) cells harbored enhanced expression of Vimentin (Figure 5A) but decreased expression of E-cadherin (Figure 5B). We also demonstrated that in the shCD47 cells, the observed reduced expression of CD47 positively correlated with marked orosphere disintegration, significant reduction in orosphere size, loss of cytoplasmic and nuclear co-localization of CD47 with Vimentin and enhanced cytoplasmic/cytomembranous co-localization of CD47 with E-cadherin (Figures 5A and 5B). Using the R2: Genomics analysis and visualization platform (https://hgserver1.amc.nl/), we carried out a bivariate analysis of the transcript expression profiles of CD47, Vimentin (VIM), or E-cadherin (CDH1) in the Roepman OSCC cohort (n = 220). Results of our analyses further validated earlier demonstrated direct and inverse correlation of CD47 with VIM and CDH1, respectively, with the significance of correlation r-value = 0.081 p-value = 0.23 T-value = 1.198 for CD47 versus VIM, and r-value = -0.061 p-value = 0.37 T-value = -0.898 for CD47 versus CDH1, with both sharing same degrees of freedom=218 (Figures 5C and 5D). In addition, for better appreciation and visualization of the place of CD47 in the interplay and functional interaction between CSCs and EMT, we generated an association network of the interaction between CD47 and molecular moieties involved in stem cell development, stem cell differentiation, stem cell maintenance, positive regulation of cell migration and positive regulation of cell motility, based on physical interactions (38.77%), genetic interactions (0.95%), co-expression (17.82%), shared protein domains (3.89%), pathway (1.67%) and molecular prediction (36.91%) (Figure 5E). These data indicate that CD47 not only interacts with, but also modulates the expression and subcellular localization of mesenchymal and epithelial factors in OSCC.
Q6: Reviewer #2: On page 4 lines 5-6 and 8, write in the correct manner the incidence of oral cancer (300, 373; 145, 353 and 242, 886 cases), remove the comma and replace it with a dot.
A1: We thank the reviewer for this comment. We already rewritten and replacing the comma with a dot changes the numerical implication or meaning of the data presented. Please see the revised Introduction section, page 4, line 2-10.
Oral cancer is one of the most diagnosed malignancies with a global annual incidence of 300,373 cases (1). Regardless of diagnostic and therapeutic progresses, from an estimated 145,353 deaths for both sexes in 2012, the oral cancer-related mortality rate continues to rise, with a predicted global mortality rate of 67.1% (n = 242,886) by 2035 because of demographic changes, thus making it one of the main causes of cancer-related deaths globally (1, 2). Constituting over 90% of all oral cancer histological sub-types, the oral squamous cell carcinoma (OSCC) is notably highly aggressive, often non-responsive to common anticancer therapy, and associated with early relapse and poor prognosis (1, 2).
Q7: Reviewer #2: Figure 2B appear much more as a Pie Chart than a Venn diagram.
A1: We thank the reviewer for this observation. We have now redraw the Pie Chart in our updated Figure 2B.
Please kindly see our Figure 2 and its legend, pages 28, line 2-9.
Figure 2. The aberrant expression of CD47 in oral squamous cell carcinoma positively correlates with disease progression. (A) Representative immunohistochemistry staining of CD47 in human OSCC tissues. (B) Pie chart showing the distribution of patients in our cohort (n=76) based on histological types. (C) Graphical representation of the histology-specific relative expression of CD47 in tissue samples from our cohort. CD47 tissue expression is relative to that in the normal group. ns, not significant; *p<0.05, **p<0.01, ***p<0.001.

Round 2
Reviewer 2 Report
I thank authors for their answers, however there are still some improvements to be made:
authors should remove the 5 samples with in situ carcinoma from the clinicopathological analysis. It's unclear if in situ carcinomas should be considered as precancerous or cancerous diseases, hence it's better to remake the survival analysis also without these samples. Authors reported: "Using the UCSC Xena functional genomics browser (https://xena.ucsc.edu/) to further characterize CD47 expression in the genome data commons (GDC) TCGA HNSC, our histomorphological stratification analyses revealed that CD47 is overexpressed in the keratinizing, non-keratinizing or ‘not otherwise specified’ (NOS) squamous cell carcinoma of the oral cavity, constituting over 80% of the HNSC and consisting of the oral tongue, buccal mucosa, alveolar, oropharynx, tonsils, floor of the mouth, and base of the tongue, compared to other histomophological types and anatomic sites." In this text it's quite clear on bioinformatics analysis authors didn't focus only on OSCC, but on the whole group of SCCHN. In my opinion, authors should download both gene-expression and clinical-data from the TCGA, then perform all the analysis only on the OSCC samples comprising survival analysis only for the OSCC patients.Author Response
Answers to the comments:
Point-by-point responses to reviewer’s comments:
We would like to thank the reviewer for the thorough reading of our manuscript as well as their valuable comments. We have followed their comments closely and feel that they have further strengthened the manuscript. Below are our point-by-point responses.
Editor Decision
after careful analysis of the opinions formulated by the reviewers regarding the new version of your manuscript I can inform you that your paper will be accepted after minor revision. In particular, the reviewer # 2 identified some aspects of histopathological analysis that need to be clarified. In particular it is reported that the authors "should remove the 5 samples with in situ carcinoma from the clinicopathological analysis. It is unclear if in situ carcinomas should be considered as precancerous or cancerous diseases, hence it's better to remake the survival analysis also without these samples". More details can be found in the reviewer's reply. Therefore, the authors are encouraged to consider these findings and to modify the manuscript accordingly. Thank you for submitting your paper to Cells.
A1: We are very grateful to the editor for the encouraging comments and more so to the reviewer for additional suggestions made. In the revised version (R4), we have tried to address the reviewer’s concern and made use of all his suggestions. The following is our point-by-point response to reviewer’s comments:
Reviewer #2
Q1. I thank authors for their answers, however there are still some improvements to be made.
A1: We thank the reviewer for taking time to review our work once again and for this encouraging comment.
Q2. authors should remove the 5 samples with in situ carcinoma from the clinicopathological analysis. It's unclear if in situ carcinomas should be considered as precancerous or cancerous diseases, hence it's better to remake the survival analysis also without these samples.
A2: We sincerely thank the reviewer for this important comment. As suggested by the reviewer, we have now removed the 5 carcinoma in situ samples from the clinicopathological analysis and redone analysis. Please see our updated Table 1, page 24, lines 3-5.
Table 1. Clinicopathological characteristics of 71 patients with OSCC.
Please see our revised Materials and Methods section, page 6, lines 2-17.
Patient Samples
We evaluated the prognostic relevance of CD47 expression in a cohort of OSCC patients (n=71) aged between 29 and 72, with median age of 48 years, who had undergone definitive treatment with curative intent in National Defense Medical Center from October 2000 to March 2013. The present study was reviewed and approved by the institutional review board (TSGHIRB 2–102-05–125), and all participants provided written informed consents. Patient distribution based on American Joint Committee on Cancer (AJCC) staging, 7th edition, was as follows - 9 stage I disease (13%), 22 stage II (31%), 13 stage III (18%), and 27 stage IV (38%), as well as normal non-tumor (2), mild dysplasia (1), moderate dysplasia (1) and severe dysplasia (3) from peritumoral tissue. Location-wise, buccal mucosa (n= 39, 51.3%) was the most common affected site, followed by tongue (n= 25, 32.9%), gingival (n= 7, 9.2%), and others (n=5, 6.6%) [palate (n= 2), maxilla (n= 1), lip (n= 1) and tonsil (n= 1)]. None of the specimens had received radiation or chemotherapeutic regimen. The clinicopathological characteristic of our cohort is shown in Table 1.
Please also see our revised Materials and Methods section, page 6, lines 19-26.
Immunohistochemical (IHC) staining
For the CD47 IHC staining analysis of OSCC (n = 71) and non-tumor oral tissues (n = 7), 1% Bovine Serum Albumin (BSA) was used for blocking the formalin-fixed paraffin-embedded (FFPE) tissue sections before they were incubated with CD47 antibody (Santa Cruz) at 4°C overnight. The sections were then incubated with goat anti-mouse IgG (Cell Signaling Technology) for 1 h. Tissue staining was scored by two independent pathologists. The staining index (SI) was calculated based on the formula.
Also kindly see our revised Results section, page 11, lines 9-22.
The aberrant expression of CD47 in human oral squamous cell carcinoma tissue positively correlates with disease progression
Furthermore, consistent with earlier data, compared to the normal or dysplastic tissues, results of our immunohistochemical staining showed varying degrees of positive CD47 staining in all 71 OSCC cases; of these 87.5% of these were membranous, 10.9% cytoplasmic, and 1.6% perinuclear staining. A strong positive correlation between enhanced CD47 protein expression and disease progression or tumor stage was established (Figure 2A). Interestingly, while we observed no apparent CD47 expression in ‘normal’ non-dysplastic tissues, we observed a graduated mild positive CD47 expression in the ‘non tumor’ mild to severely dysplastic tissues, moderate expression of CD47 in the early stage (I, II) carcinoma (p<0.05 vs. normal or mild dysplasia), and strong CD47 staining in the late stage (III and IV) group (p<0.001 vs. normal or mild dysplasia), especially in the cytomembranous region (Figure 2A-C).
Q3. Authors reported: "Using the UCSC Xena functional genomics browser (https://xena.ucsc.edu/) to further characterize CD47 expression in the genome data commons (GDC) TCGA HNSC, our histomorphological stratification analyses revealed that CD47 is overexpressed in the keratinizing, non-keratinizing or ‘not otherwise specified’ (NOS) squamous cell carcinoma of the oral cavity, constituting over 80% of the HNSC and consisting of the oral tongue, buccal mucosa, alveolar, oropharynx, tonsils, floor of the mouth, and base of the tongue, compared to other histomophological types and anatomic sites." In this text it's quite clear on bioinformatics analysis authors didn't focus only on OSCC, but on the whole group of SCCHN. In my opinion, authors should download both gene-expression and clinical-data from the TCGA, then perform all the analysis only on the OSCC samples comprising survival analysis only for the OSCC patients
A3: We sincerely appreciate the reviewer’s comment. To better addressing the reviewer’s concern, we have now rephrased the paragraph for clarity and better comprehensibility in our revised manuscript. Please kindly see our Results section, Pages 10, line 21-41 to page 11, line 1-7.
CD47 is aberrantly expressed in human oral squamous cell carcinoma and influence survival rate
To understand the role of CD47 in OSCC, we performed computational analyses of CD47 RNAseq expression profile in 33 different cancer types (n = 9,736 tumors) including head and neck squamous cell carcinoma (HNSC) matched with corresponding normal samples from the cancer genome atlas (TCGA) and genotype-tissue expression (GTEx) (n = 8,587 non-tumors) datasets, using the analysis of variance for evaluation of differential expression. Results of these analyses indicate that CD47 was overexpressed in tumors compared to adjacent non-tumor oral tissues and was mostly expressed in HNSC tissues with ~ 125 transcripts per million (TPM) after the ovarian (OV, ~198 TPM) and lung adenocarcinoma (LUAD, ~195 TPM) (Figure 1A and Supplementary Figure S1). Furthermore, statistical analyses of OSCC cohort data (consisting of the oral tongue, buccal mucosa, alveolar, oropharynx, tonsils, floor of the mouth, and base of the tongue) downloaded from the UCSC Xena functional genomics browser (https://xena.ucsc.edu/) was used to further characterize CD47 expression in patients with OSCC; our results indicate that CD47 is overexpressed in the keratinizing, non-keratinizing and ‘not otherwise specified’ (NOS) squamous cell carcinoma of the oral cavity, which constitutes over 75% of HNSC (Figure 1B). Further computational analyses of OSCC in TCGA cohort (n=412; p = 0.0009) showed that compared to expression in the ‘normal’ oral epithelium (n = 32), CD47 was significantly more expressed in the OSCC tissue samples (n = 380) (Figure 1C).
We also demonstrated using downloaded and reanalyzed malignant OSCC data from the TCGA HNSC cohort, that high CD47 expression confers a significant survival disadvantage in OSCC patients with high CD47 expression, compared to those with low CD47 expression (p = 0.0391; Figure 1D).
Also kindly see our updated Figure 1 and its legend, page 26, line 3-11.
Figure 1. CD47 is aberrantly expressed in human oral squamous cell carcinoma and influence survival rate. (A) CD47 transcript expression profile across TCGA and GTEx paired normal-tumor tissue cohort. (B) The expression of CD47 in downloaded data for OSCC based on morphology, anatomic site, and sample type from the GDC TGCA HNSC dataset. (C) Differential expression of CD47 in normal oral and cancer tissues in TCGA OSCC cohort (n=412; p = 0.0009). (D) Kaplan-Meier curves showing the effect of low and high CD47 expression on the overall survival of TGCA malignant OSCC cohort.

This manuscript is a resubmission of an earlier submission. The following is a list of the peer review reports and author responses from that submission.
Round 1
Reviewer 1 Report
In this manuscript the authors investigate the role of CD47, a controller of phagocytosis and immune-regulation, in patient samples and in-vitro models of oral carcinoma. The authors claim that the expression of CD47 correlates with disease progression and CD47 supports cancer stem cells maintenance and EMT. The paper is well written and experiments well planned. The usage of human samples and the variety of approaches used increases the value of the paper. Overall, the manuscript raises interesting questions for the field and could benefit from the addition of a few additional experiments and points of discussion:
Major points:
1. How CD47 regulates cancer stem cells maintenance? Is it increasing their proliferation, survival or preventing differentiation? The authors should provide additional experimental insights and discuss their point of view.
2. Are undifferentiated cells responsible for increased EMT or the two effects are independent from each other? Are the CSC more motile because more immature? The authors should provide scientific evidence and discuss this point in more details.
3. The authors should increase, if possible, the size of patient samples (it is a little limited).
Minor points:
1. Labelling and resolution of figures should be improved
2. Sentence in lines 566-571 should be rephrased to improve clarity
3. Supplementary figure 3 is slightly distorted
4. References should contain up to 10th authors before “et al.”
Author Response
Answers to the comments:
Point-by-point responses to reviewer’s comments:
We would like to thank the reviewer for the thorough reading of our manuscript as well as their valuable comments. We have followed their comments closely and feel that they have further strengthened the manuscript. Below are our point-by-point responses.
Q1: Reviewer #1: This study highlights the important role of CD47 in the pathogenesis of OSCCs. The data is well-presented and overall findings novel. One of the weakest points of this work is the lack of association made with human papilloma viruses (HPVs) in OSCCs, a major culprit in over 1/3 of OSCCs. Nowhere was HPV even mentioned by name in the manuscript despite the fact that the incidence of HPV-positive HNSCCs have recently exceeded that of HPV-driven cervical cancer (Morbidity and Mortality Weekly Report (MMWR) CDCMMWR Trends in Human Papillomavirus–Associated Cancers — United States, 1999–2015 Weekly / August 24, 2018 / 67(33);918–924) making HPV-positive HNSCCs the next major HPV-driven cancer type. Authors are strongly asked to discuss their findings in light of HPV, if possible. Nonetheless, work presented is ample for publication in the journal and authors are therefore asked to address some of these comments below:
A1: We thank the reviewer for this important comment. We do agree that there is a strong relationship between human papilloma virus (HPV) infection and OSCC. We have now discussed this relationship briefly in our revised manuscript. Please kindly see Lines 584-607.
Human papilloma virus (HPV) infection has been implicated in over 25% of HNSCCs. In fact, many of the HPV positive HNSCCs are oropharynx cancer including tongue base and tonsil (35, 36). Clinically, the HPV status of patients is being touted as a predictor of treatment response and survival rate (37, 38). It has been suggested that the better prognosis of HPV(+) HNSCCs involves the host immune system, especially as immune cells, which were originally suppressed in the tumor microenvironment, were shown to be re-activated through the crosstalk of immunogenic signals between tumorous cells and immune cells by exosomes (38). Tumor released exosomes containing a lot of antigens. In contrast to HPV(-) tumor, exosomes released by HPV (+) tumors do exhibit prolong immunogenic interaction with immune cells due to the regulation of the expression and/or activity of CD47 on the membrane of cancerous cells. The membrane protein CD47 functions as “don’t eat me” signal and thus limit the clearance of cancerous cells by circulating monocytes (38, 39). Radiation has been shown to induce anti-tumor immune response (40) and associated with dose-dependent decrease in the surface expression of CD47 on HPV(+) HNSCC cells, resulting in improved clearance of tumorous cell, in vitro and in vivo (39).
Our initial analysis of the TCGA-HNSCC cohort (n = 604) gene expression profile dataset showed it contained limited number of the HPV positive HNSCC as sorted by FISH or p16 testing; most of the tumor sites are located at the tongue base and oropharynx, which were excluded in our original TCGA analysis since our study focus was oral cancer instead. In addition, the expression level of CD47 in these samples was low or null. These findings suggest that the therapeutic potential of targeting CD47 is likely independent of the HPV status of the patients with OSCC. The mechanism underlying the response of CD47 to radiation in OSCC remains relatively underexplored, thus, necessitating further investigation of the use CD47 as potential therapeutic target in anti-OSCC treatment.
Please kindly see our revised Reference Section, Lines 729-748.
Gillison ML, Koch WM, Capone RB, Spafford M, Westra WH, Wu L, Zahurak ML, Daniel RW, Viglione M, Symer DE, et al. Evidence for a causal association between human papillomavirus and a subset of head and neck cancers. J Natl Cancer Inst. 2000;92 (9):709–720. van Houten VM, Snijders PJ, van Den Brekel MW, Kummer JA, Meijer CJ, van Leeuwen B, Denkers F, Smeele LE, Snow GB, Brakenhoff RH. Biological evidence that human papillomaviruses are etiologically involved in a subgroup of head and neck squamous cell carcinomas. Int J Cancer. 2001;93(2):232–235. Doi:10.1002/ijc.1313. Ou D, Blanchard P, El Khoury C, De Felice F, Even C, Levy A, Nguyen F, Janot F, Gorphe P, Deutsch E, et al. Induction chemotherapy with docetaxel, cisplatin and fluorouracil followed by concurrent chemoradiotherapy or chemoradiotherapy alone in locally advanced non-endemic nasopharyngeal carcinoma.Oral Oncol. 2016; 62:114-121. Ludwig S, Marczak L, Sharma P, Abramowicz A, Gawin M, Widlak P, Whiteside TL, Pietrowska M. Proteomes of exosomes from HPV(+) or HPV(-) head and neck cancer cells: differential enrichment in immunoregulatory proteins. Oncoimmunology. 2019; 8(7):1593808. doi: 10.1080/2162402X.2019.1593808. Vermeer DW, Spanos WC, Vermeer PD, Bruns AM, Lee KM, Lee JH. Radiation-induced loss of cell surface CD47 enhances immune-mediated clearance of human papillomavirus-positive cancer. Int J Cancer. 2013 Jul;133(1):120-9. doi: 10.1002/ijc.28015. Golden EB, Apetoh L. Radiotherapy and immunogenic cell death. Semin Radiat Oncol. 2015 Jan;25(1):11-7. Doi: 10.1016/j.semradonc.2014.07.005.
Q2: Reviewer #1: Pg 10, Line 347 – Did you mean to say immunofluorescence, instead of IHC?.
A2: We thank the reviewer for this important observation. As rightly pointed oit by the reviewer, we intended to write immunofluorescence staining, and we have made the correction in our revised manuscript.
Please kindly see Lines 347-351.
These Western blot results were corroborated by immunofluorescence (IFC) staining showing concurrent remarkable reduction in orosphere size and expression levels of CD47, OCT4, c-MYC, and SOX2 protein in the shCD47-1 and shCD47-2 TW2.6 cells, compared with their WT counterpart (Figure 3D).
Q3: Reviewer #1: Figure 3B-D – The figure legend failed to state the number of replicate experiments done here. Please mention this Supp fig 3A and B fig legend.
A2: We appreciate the reviewer’s suggestion. This is already indicated in the Statistical Analysis section, please kindly see Line 258-264.
Statistical Analysis
Each experiment was performed at least 3 times in triplicates. All statistical analyses were carried out using IBM SPSS Statistics for Windows, Version 25.0 (Released 2017; Armonk, NY: IBM Corp. USA). All data represent means ± standard deviation (SD). Comparison between two groups was estimated using the 2-sided Student's t-test, while the one-way analysis of variance (ANOVA) was used for comparison between 3 or more groups. P-value <0.05 was considered statistically significant.
However, to allay the reviewer’s concern, we have now included this in the revised manuscript. Please kindly see Lines 355-366.
Figure 3. CD47 modulates the cancer stem cell-like phenotype in oral squamous cell carcinoma cells. (A) Box and whiskers chart showing the correlative differential expression of CD47 (upper), SOX2 (middle) and CD133 (lower) mRNA from analyses of the Human OSCC Genome U133A Array from the Toruner Head-Neck cohort, n=20; (B) Knockdown efficiency of shCD47-1 and shCD47-2 on the protein expression of CD47 in SAS and TW2.6 cell lines shown by western blot analysis. (C) Effect of CD47 knockdown on the expression level of CD47, Sox2, Oct4, and CD133 proteins in SAS Sp, shCD47-1 or shCD47-2 cells shown by western blot analysis. GAPDH served as loading control. (D) Immunofluorescent staining showing the effect of shCD47 on the expression of CD47, Oct4, c-Myc and Sox2 proteins in spheres formed by TW 2.6 cells. All assays are representative of experiments performed 4 times in triplicates. WT, wild type; Sp, orosphere; blue stain = DAPI, nuclear staining.
Q4: Reviewer #1: IC50s are only done for drug concentrations not radiation doses. Even if it is proper analysis for determining IC50 was not shared here. Authors are asked to remove any mere mention of this.
A2: We really thank the reviewer for this very important comment. As suggested by the reviewer, we have now removed any mere mention of IC50 with regards to radiation. Please kindly see Line 436-468.
Suppression of CD47 expression enhances the sensitivity of OSCC-SCs to radiation therapy
Understanding the critical role of radiotherapy as a vital treatment modality in OSCC that facilitates oral tumor size reduction and oral function preservation, we next examined the effect of altered CD47 expression on the viability and/or proliferation of the OSCC cell lines, SAS and TW2.6 using the radiotherapy-based cell viability assay. 24 h post-transfection with shCD47-1, shCD47-2 or negative control scramble shCD47, SAS cells were exposed to 5 Gy – 15 Gy radiation. The combination of shCD47 and irradiation induced significantly greater cell-death in comparison to the radiation only group, as evidenced by ~52% vs 46%, 24% vs 19%, and 28% vs 20% reduction in the viability of shCD47-1 vs shCD47-2 transfected SAS cells treated with 5 Gy, 10 Gy, and 15 Gy, respectively (Figures 6A and 6B, also see Supplementary Figures S3A and 3B). As shown in Figure 6B, irradiation reduced the cell viability by 50% at 8.59 Gy, 1.06 Gy and 1.71 Gy for WT, shCD47-1 and shCD47-2 SAS cells, respectively. Along same line, we evaluated the probable enhanced effect of combining molecular attenuation of CD47 with low-dose radiotherapy. Our results showed that compared to migration of the un-irradiated SAS WT, significantly lesser migratory potential was exhibited by the 5 Gy irradiated shCD47-1 and shCD47-2 transfected cells (~5.2-fold, p<0.001), un-irradiated shCD47 alone (~2.7-fold, p<0.001), and 5 Gy radiation alone SAS WT (1.8-fold, p<0.01) (Figure 6C). Similarly, a 3.5-fold (p<0.01), 3.0-fold (p<0.01), or 12.0-fold (p<0.001) reduction in the number of invaded un-irradiated shCD47-transfected, 5 Gy radiation alone, or 5 Gy irradiated shCD47-transfected cells, respectively (Figure 6D and Supplementary Figure S3C). Consistent with the CD47-CSCs-EMT positive feedback loop already alluded to above, we also demonstrated that compared to the orospheres generated from SAS WT shCD47 cells, fewer and smaller orospheres were formed by SAS or HSC-3 WT cells irradiated with 5 Gy alone, cells transfected with shCD47-1 alone, and 5 Gy-irradiated cells harboring shCD47-1, in decreasing order of magnitude (Figure 6E and Supplementary Figure S3D). Furthermore, in combination with shCD47, exposure to 5 Gy radiation reduced the number of colonies formed by ~8.69 - 11.1-fold (p<0.001) in comparison to un-irradiated SAS WT (Figure 6F, also see Supplementary Figure S3E). Our data does indicate that the observed resultant enhanced anticancer effect of radiation therapy combined with shCD47 in OSCC -SCs is primarily synergistic and highly effective. These results demonstrate that molecular attenuation of shCD47 as a therapeutic strategy abrogates the CSCs-related radio-resistance of OSCC cells.

Reviewer 2 Report
This study highlights the important role of CD47 in the pathogenesis of OSCCs. The data is well-presented and overall findings novel. One of the weakest points of this work is the lack of association made with human papilloma viruses (HPVs) in OSCCs, a major culprit in over 1/3 of OSCCs. Nowhere was HPV even mentioned by name in the manuscript despite the fact that the incidence of HPV-positive HNSCCs have recently exceeded that of HPV-driven cervical cancer (Morbidity and Mortality Weekly Report (MMWR) CDCMMWR Trends in Human Papillomavirus–Associated Cancers — United States, 1999–2015 Weekly / August 24, 2018 / 67(33);918–924) making HPV-positive HNSCCs the next major HPV-driven cancer type. Authors are strongly asked to discuss their findings in light of HPV, if possible. Nonetheless, work presented is ample for publication in the journal and authors are therefore asked to address some of these comments below:
Pg 10, Line 347 – Did you mean to say immunofluorescence, instead of IHC? Figure 3B-D – The figure legend failed to state the number of replicate experiments done here. Please mention this Supp fig 3A and B fig legend – IC50s are only done for drug concentrations not radiation doses. Even if it is proper analysis for determining IC50 was not shared here. Authors are asked to remove any mere mention of thisAuthor Response
Q1: Reviewer #2: In this manuscript the authors investigate the role of CD47, a controller of phagocytosis and immune-regulation, in patient samples and in-vitro models of oral carcinoma. The authors claim that the expression of CD47 correlates with disease progression and CD47 supports cancer stem cells maintenance and EMT. The paper is well written and experiments well planned. The usage of human samples and the variety of approaches used increases the value of the paper. Overall, the manuscript raises interesting questions for the field and could benefit from the addition of a few additional experiments and points of discussion.
A2: We thank the reviewer for all the comments and suggestions made. We find them non-prejudicial and helpful. We have revised our manuscript once again based on these comments and do hope we have now addressed all the reviewer’s concerns and now meet the threshold for acceptance.
Q2: Reviewer #2: How CD47 regulates cancer stem cells maintenance? Is it increasing their proliferation, survival or preventing differentiation? The authors should provide additional experimental insights and discuss their point of view.
A2: We thank the reviewer for this important question. While the present study demonstrates a regulatory link between CD47 expression level and acquisition and/or maintenance of stem cell-like phenotypes, the probable underlying mechanism remains unclear and is the subject of on-going studies. Nevertheless, we humbly refer the reviewer to our Discussion section, Lines 549-583.
Aside demonstrating that CD47 modulates the expression and subcellular localization of mesenchymal and epithelial factors in OSCC, we also provided evidence, at least in part, that CD47 is a master regulator of stem cell development, differentiation, and maintenance, as well as positive regulator of cell migration and cell motility (Figure 5). These findings are particularly insightful and therapeutically-relevant, especially as the “migrating, self-renewing and symmetrically-dividing CSCs shape the primary tumor, and are also exclusively capable of distant seeding, whereas the majority of non-stem cancer cells (that can be frequently detected as circulating tumor cells) are intrinsically only able to form dormant micrometastases” (27). To the best of our understanding of the OSCC SC-EMT, while it is difficult to delineate clearly whether it is the undifferentiated or differentiated OSCC cells are responsible for observed EMT phenotype in our study, we would allude to the recent comprehensive overview of related theme, in which Kim et al. suggested that “early, undifferentiated cells with mesenchymal phenotype are characterized by a shift from E-cadherin expression to N-cadherin expression along with the expression of Snails, vimentin and metalloproteases”, and that these “early undifferentiated cells with a mesenchymal phenotype retain the expression of several totipotent transcription factors (e.g., Oct4 and Nanog), which indicates that these cells can adopt a mesenchymal phenotype without losing their pluripotency” (32). In the light of this, we posit a regulatory role for CD47 signaling at the interphase between the CSC and EMT phenotype of OSCC cells.
By inference, extrinsic perturbations such as CD47 blockage/knockdown and cytotoxic treatments including radiation therapy may mold oral tumor by selective targeting of the aggressive OSCC cells, including OSCC-SCs which subsequently facilitate malignant growth; thus, any efficacious therapy must eradicate OSCC-SCs, however, there is mounting evidence showing these cells are intrinsically less or in-sensitive to current OSCC anticancer therapy. Thus, interestingly, having shown the existence of a positive correlation of CD47 expression with CSC, EMT and metastatic phenotypes, as well as an inverse correlation with OSCC patients’ overall survival, cell death; we finally demonstrated that the suppression of CD47 expression enhances the sensitivity of OSCC-SCs to radiation therapy, as evidenced by the remarkable synergistic effect of concurrent CD47 knockdown and radiotherapy on cell viability, migration, invasiveness, clonogenic and orospheric survival (Figure 6). This has clinical significance since radiotherapy is a common and very vital component of the multidisciplinary treatment for patients with OSCC, especially those with unresectable oral cavity tumors, cases where surgery is technically-improbable early stage disease with high risk of cosmetic or functional defect, high operative risk secondary to co-morbidity or subpar performance status (33, 34).
Also kindly see Lines 400-424.
CD47 modulates the expression and subcellular localization of mesenchymal and epithelial factors in OSCC
In parallel assays, immunofluorescence (IFC) staining showed that compared to expression in the negative control shCD47 scramble cells, the SAS orospheres (SAS Sp) cells harbored enhanced expression of Vimentin (Figure 5A) but decreased expression of E-cadherin (Figure 5B). We also demonstrated that in the shCD47 cells, the observed reduced expression of CD47 positively correlated with marked orosphere disintegration, significant reduction in orosphere size, loss of cytoplasmic and nuclear co-localization of CD47 with Vimentin and enhanced cytoplasmic/cytomembranous co-localization of CD47 with E-cadherin (Figures 5A and 5B). Using the R2: Genomics analysis and visualization platform (https://hgserver1.amc.nl/), we carried out a bivariate analyses of the transcript expression profiles of CD47, Vimentin (VIM), or E-cadherin (CDH1) in the Roepman OSCC cohort (n = 220). Results of our analyses further validated earlier demonstrated direct and inverse correlation of CD47 with VIM and CDH1, respectively, with the significance of correlation r-value = 0.081 p-value = 0.23 T-value = 1.198 for CD47 versus VIM, and r-value = -0.061 p-value = 0.37 T-value = -0.898 for CD47 versus CDH1, with both sharing same degrees of freedom=218 (Figures 5C and 5D). In addition, for better appreciation and visualization of the place of CD47 in the interplay and functional interaction between CSCs and EMT, we generated an association network of the interaction between CD47 and molecular moieties involved in stem cell development, stem cell differentiation, stem cell maintenance, positive regulation of cell migration and positive regulation of cell motility, based on physical interactions (38.77%), genetic interactions (0.95%), co-expression (17.82%), shared protein domains (3.89%), pathway (1.67%) and molecular prediction (36.91%) (Figure 5E). These data indicate that CD47 not only interacts with, but also modulates the expression and subcellular localization of mesenchymal and epithelial factors in OSCC.
Q3: Reviewer #2: Are undifferentiated cells responsible for increased EMT or the two effects are independent from each other? Are the CSC more motile because more immature? The authors should provide scientific evidence and discuss this point in more details.
A2: We thank the reviewer for these questions. While these questions are fundamentally relevant to understanding the EMT-CSC association, answers to them are divergent. To some extent we alluded to contemporary understanding of these theme as laid out in published works referenced in our manuscript. Please kindly see Lines91-100.
Corollary to the above implication of CSCs in cancer metastases and resistance to anticancer therapy, as well as accruing association of the aberrant expression of epithelial-to-mesenchymal transition (EMT)-inducing transcription factors with cancer stemness, the last decade has been characterized by increased documentation of existent reciprocal link between CSCs and EMT; with CSCs exhibiting EMT phenotypes, and EMT being relevant to the acquisition and maintenance of stem cell-like traits and sufficient to confer same traits on differentiated non-cancerous and cancerous cells (10 – 12). This CSCs-EMT reciprocity, mirroring a pathological positive feed-back loop plays a critical role in enhanced chemoresistance, radiotherapy resistance, disease progression, recurrence and poor prognosis (10 – 13).
Please kindly see our revised Reference Section, Lines 665-673.
Liu X, Fan D. The epithelial-mesenchymal transition and cancer stem cells: functional and mechanistic links. Curr Pharm Des. 2015; 21(10): 1279-91. doi: 2174/1381612821666141211115611 Sato R, Semba T, Saya H, Arima Y. Concise Review: Stem Cells and Epithelial-Mesenchymal Transition in Cancer: Biological Implications and Therapeutic Targets. Stem Cells. 2016; 34(8): 1997-2007. doi: 10.1002/stem.2406 Ye X, Weinberg RA. Epithelial-Mesenchymal Plasticity: A Central Regulator of Cancer Progression. Trends Cell Biol. 2015; 25(11): 675-686. doi: 10.1016/j.tcb.2015.07.012. Shibue T, Weinberg RA. EMT, CSCs, and drug resistance: the mechanistic link and clinical implications. Nat Rev Clin Oncol. 2017;14(10):611–629. doi:10.1038/nrclinonc.2017.44
Please kindly see our revised Discussion section, Lines 557-567.
To the best of our understanding of the OSCC SC-EMT, while it is difficult to delineate clearly whether it is the undifferentiated or differentiated OSCC cells are responsible for observed EMT phenotype in our study, we would allude to the recent comprehensive overview of related theme, in which Kim et al. suggested that “early, undifferentiated cells with mesenchymal phenotype are characterized by a shift from E-cadherin expression to N-cadherin expression along with the expression of Snails, vimentin and metalloproteases”, and that these “early undifferentiated cells with a mesenchymal phenotype retain the expression of several totipotent transcription factors (e.g., Oct4 and Nanog), which indicates that these cells can adopt a mesenchymal phenotype without losing their pluripotency” (32). In the light of this, we posit a regulatory role for CD47 signaling at the interphase between the CSC and EMT phenotype of OSCC cells.
Please kindly see our revised Reference Section, Lines 720-722.
Kim DH, Xing T, Yang Z, Dudek R, Lu Q, Chen YH. Epithelial Mesenchymal Transition in Embryonic Development, Tissue Repair and Cancer: A Comprehensive Overview. J Clin Med. 2017 Dec 22;7(1):1. doi: 10.3390/jcm7010001. PubMed PMID: 29271928; PubMed Central PMCID: PMC5791009.
Q4: Reviewer #2: The authors should increase, if possible, the size of patient samples (it is a little limited).
A2: We thank the authors for this suggestion. While we would have loved to do this, mitigating factors such as incomplete clinicopathological data for a good number of our OSCC cohort as well as the challenge of obtaining new informed consents from new subjects limits us.
Q5: Reviewer #2: Minor points: 1. Labelling and resolution of figures should be improved
A2: We thank the authors for this suggestion. We thank the reviewer for this suggestion. We have addressed this in our revised manuscript with updated figures. Kindly see our updated Figure section.
Q6: Reviewer #2: Sentence in lines 566-571 should be rephrased to improve clarity.
A2: We are grateful the reviewer pointed this out. We have now corrected this paragraph in our revised manuscript. Please kindly see revised discussion section, Lines 568-583.
By inference, extrinsic perturbations such as CD47 blockage/knockdown and cytotoxic treatments including radiation therapy may mold oral tumor by selective targeting of the aggressive OSCC cells, including OSCC-SCs which subsequently facilitate malignant growth; thus, any efficacious therapy must eradicate OSCC-SCs, however, there is mounting evidence showing these cells are intrinsically less or in-sensitive to current OSCC anticancer therapy. Thus, interestingly, having shown the existence of a positive correlation of CD47 expression with CSC, EMT and metastatic phenotypes, as well as an inverse correlation with OSCC patients’ overall survival, cell death; we finally demonstrated that the suppression of CD47 expression enhances the sensitivity of OSCC-SCs to radiation therapy, as evidenced by the remarkable synergistic effect of concurrent CD47 knockdown and radiotherapy on cell viability, migration, invasiveness, clonogenic and orospheric survival (Figure 6). This has clinical significance since radiotherapy is a common and very vital component of the multidisciplinary treatment for patients with OSCC, especially those with unresectable oral cavity tumors, cases where surgery is technically-improbable early stage disease with high risk of cosmetic or functional defect, high operative risk secondary to co-morbidity or subpar performance status (33, 34).
Q7: Reviewer #2: Supplementary figure 3 is slightly distorted.
A2: We appreciate the reviewer’s observation. We have addressed this in our revised manuscript.
Q8: Reviewer #2: References should contain up to 10th authors before “et al.”
A2: We thank the reviewer for drawing our attention to this. We have now addressed this in our revised submission. Kindly see our revised Reference section, Lines 639-748.

Round 2
Reviewer 1 Report
The authors have provided replies to some of the concerns described in the 1st reviewing process.
However, points of importance are not touched. more specifically:
Major points:
- The control of stemness via CD47 should be regarded as essential element in the analysis, as it can provide important insight in the understanding of the cancer recurrence and sustain of growth. In the new version of the manuscript the authors discussed the point in more details for what concern transcriptional changes, size of spheres and colonies formation. But they did not provide a conclusive self-renewal assay. Classically, this is done by seeding a single cell/per well and let it grow until full mature sphere. The sphere will then be dissociated and re-plated at single cell dilution, to prove that it can grow again. Only stem cells will be able to self-renew for several passages. Can the authors perform this experiment in control (CD47+) and shCD47-treated conditions? Is the read-out showing differences of self-renewal (i.e. number and size of newly formed spheres from single cells through several passages)? What is the % of single cells giving rise to spheres in the two conditions? Does this number change increasing the number of passages?
- We understand the difficulty and limitation of gathering additional human samples.
Minor points:
-The authors addressed and corrected previous concerns.
Author Response
Answers to the comments:
Point-by-point responses to reviewer’s comments:
We would like to thank the reviewer for the thorough reading of our manuscript as well as their valuable comments. We have followed their comments closely and feel that they have further strengthened the manuscript. Below are our point-by-point responses.
Q1: Reviewer #1: The authors have provided replies to some of the concerns described in the 1st reviewing process.
However, points of importance are not touched. more specifically:
Major points:
- The control of stemness via CD47 should be regarded as essential element in the analysis, as it can provide important insight in the understanding of the cancer recurrence and sustain of growth. In the new version of the manuscript the authors discussed the point in more details for what concern transcriptional changes, size of spheres and colonies formation. But they did not provide a conclusive self-renewal assay. Classically, this is done by seeding a single cell/per well and let it grow until full mature sphere. The sphere will then be dissociated and re-plated at single cell dilution, to prove that it can grow again. Only stem cells will be able to self-renew for several passages. Can the authors perform this experiment in control (CD47+) and shCD47-treated conditions? Is the read-out showing differences of self-renewal (i.e. number and size of newly formed spheres from single cells through several passages)? What is the % of single cells giving rise to spheres in the two conditions? Does this number change increase the number of passages?
- We understand the difficulty and limitation of gathering additional human samples.
Minor points: -The authors addressed and corrected previous concerns.:
A1: We thank the reviewer for taking time to read through our paper again, providing this important comment and giving us this rare second chance to address his initial concerns. As noted above, we sincerely apologize for what seemed like merely intervening correctly on the text, acknowledging the comments made to us but being unable or unwilling to respond to the request for some further experiments necessary to clarify some important issues in our initial revision. We have now included these data in Updated Figure 3 as rightly requested again by the reviewer.
Please kindly see our revised Results section, Lines 342-377.
CD47 modulates the cancer stem cell-like phenotype and self-renewal in oral squamous cell carcinoma cells
Having established a correlation between OSCC carcinogenesis, poor prognosis and CD47 expression, we further sought to unravel the underlying mechanism. Against the background that c-Myc is a master regulator of cancer transcriptome, epigenome and immune privilege (16, 17), and that OSCC SCs are characterized by ectopic expression of CD133, and pluripotency master regulators including SOX2, OCT4, and NANOG (18, 19), we probed for probable correlation and/or functional association between CD47, OCT4, SOX2, CD133, and c-MYC. Preliminary analyses of the Human OSCC Genome U133A Array of the Toruner HNSC cohort (n=20) showed concomitant upregulation of CD47 (1.55-fold, p = 0.00003), SOX2 (1.40-fold, p = 0.01) and CD133 (1.13-fold, p = 0.06) in the OSCC tissues, compared to normal tissue group (Figure 3A). Further, after shRNA-mediated silencing of CD47 expression generating shCD47-1 and shCD47-2 in SAS (knockdown efficiency: 92% and 90%, respectively) and TW2.6 (knockdown efficiency: 93% and 80%, respectively) cell lines (Figure 3B). In similar experiments, our western blot analyses revealed that shCD47 was elicited significant co-suppression of CD47, SOX2, OCT4, and CD133 in orosphere-derived SAS, HSC-3 and FaDu cells transfected with shCD47 (Figure 3C and Supplementary Figure S2). These Western blot results were corroborated by immunofluorescence (IFC) staining showing concurrent remarkable reduction in orosphere size and expression levels of CD47, OCT4, c-MYC, and SOX2 protein in the shCD47-1 and shCD47-2 TW2.6 cells, compared with their WT counterpart (Figure 3D). These results are suggestive of a functional association between CD47 and CSCs regulators in OSCC cells, as well as highlight the probable modulatory role of CD47 on the CSCs-like phenotype of OSCC cells. However, understanding that at the core of OSCC chemoresistance, metastasis, and recurrence lies the capability of CSCs to regenerate all components of the primary or original tumor and drive cancer aggression (6, 7, 10-13), we examined if and how shCD47 affects the orosphere-forming capability of single cell solutions derived from dissociated primary orospheres and cultured in stem cell media (recapitulating OSCC-SC self-renewal). We demonstrated that shCD47-1 and shCD47-2 markedly inhibited the self-renewal potential of the SAS (shCD47-1: 69.1% inhibition, p < 0.001; shCD47-2: 88.3% inhibition, p < 0.001) or TW2.6 (shCD47-1: 71% inhibition, p < 0.05; shCD47-2: 73.5% inhibition, p < 0.01) primary orospheres (Figures 3E and 3F). In addition, similar to the primary orospheres, we showed that the shCD47-induced reduction in the number and sizes of SAS or TW2.6 orospheres positively correlated with significant inhibition of the nuclear expression of pluripotency transcription factors, SOX2 and OCT4 (Figure 3G). These data indicate that shCD47 efficaciously suppresses the CSC-like phenotype, including the expansion and self-renewal of OSCC-SC population, in vitro.
Please kindly see our revised Figure 3 and its legend, Lines380-393.
Figure 3. CD47 modulates the cancer stem cell-like phenotype in oral squamous cell carcinoma cells. (A) Box and whiskers chart showing the correlative differential expression of CD47 (upper), SOX2 (middle) and CD133 (lower) mRNA from analyses of the Human OSCC Genome U133A Array from the Toruner Head-Neck cohort, n=20; (B) Knockdown efficiency of shCD47-1 and shCD47-2 on the protein expression of CD47 in SAS and TW2.6 cell lines shown by western blot analysis. (C) Effect of CD47 knockdown on the expression level of CD47, Sox2, Oct4, and CD133 proteins in SAS Sp, shCD47-1 or shCD47-2 cells shown by western blot analysis. GAPDH served as loading control. (D) Immunofluorescent staining showing the effect of shCD47 on the expression of CD47, Oct4, c-Myc and Sox2 proteins in spheres formed by TW 2.6 cells. TW2.6 and SAS cells transfected with shCD47-1 or shCD47-2 exhibited decreased orosphere size (left) and number (right) in both (E) primary and (F) secondary generation orospheres. (G) shCD47 attenuated OCT4 and SOX2 expression and inhibited their nuclear co-localization in TW2.6- or SAS- derived orospheres as shown by immunofluorescent (IFC) staining. All assays are representative of experiments performed 4 times in triplicates. WT, wild type; Sp, orosphere; blue stain = DAPI, nuclear staining.
Also kindly see our revised Materials & Methods section, Lines 182-195.
Orosphere Formation and Self-Renewal Assay
For generation of orospheres, the OSCC cells were seeded at 5 x 104 cells/well in 6-well non-adherent plates (Corning Inc., Corning, NY) in serum-free DMEM/F12 medium (ThermoFisher Scientific, 11330057) supplemented with bFGF (20 ng/mL, Invitrogen, Carlsbad, CA), B27 supplement (Invitrogen, Carlsbad, CA), 5 μg/mL insulin (Sigma-Aldrich, 91077C) and EGF (20 ng/mL, Millipore, Bedford, MA). The cells were then incubated at 37oC in 5% humidified CO2 incubator for 12-15 days, and the formed orospheres were observed and counted using inverted phase contrast microscope. After 12 days of culture, primary orospheres consisting of ≥50mm was counted, and images taken under microscope. Secondary orospheres were generated by dissociating primary orospheres using trypsinization method, dissociated primary orospheres were then pippetted through a 22G needle to obtain a single-cell suspension (Thermo Fisher Scientific Inc.). After dissociation of the primary orospheres, cell seeding was done as for primary tumorspheres. After 12-15 days of culture, secondary orospheres consisting of ≥50mm was counted, and images taken under microscope.
